# Distribution-Free Model-Agnostic Regression Calibration via Nonparametric Methods

**Shang Liu**[*]
Imperial College Business School
Imperial College London
s.liu21@imperial.ac.uk

**Zhongze Cai**[*]
Imperial College Business School
Imperial College London
z.cai22@imperial.ac.uk

**Xiaocheng Li**
Imperial College Business School
Imperial College London
xiaocheng.li@imperial.ac.uk

## Abstract

In this paper, we consider the uncertainty quantification problem for regression models. Specifically, we consider an individual calibration objective for characterizing the quantiles of the prediction model. While such an objective is well-motivated from downstream tasks such as newsvendor cost, the existing methods have been largely heuristic and lack of statistical guarantee in terms of individual calibration. We show via simple examples that the existing methods focusing on population-level calibration guarantees such as average calibration or sharpness can lead to harmful and unexpected results. We propose simple nonparametric calibration methods that are agnostic of the underlying prediction model and enjoy both computational efficiency and statistical consistency. Our approach enables a better understanding of the possibility of individual calibration, and we establish matching upper and lower bounds for the calibration error of our proposed methods. Technically, our analysis combines the nonparametric analysis with a covering number argument for parametric analysis, which advances the existing theoretical analyses in the literature of nonparametric density estimation and quantile bandit problems. Importantly, the nonparametric perspective sheds new theoretical insights into regression calibration in terms of the curse of dimensionality and reconciles the existing results on the impossibility of individual calibration. To our knowledge, we make the first effort to reach both individual calibration and finite-sample guarantee with minimal assumptions in terms of conformal prediction. Numerical experiments show the advantage of such a simple approach under various metrics, and also under covariates shift. We hope our work provides a simple benchmark and a starting point of theoretical ground for future research on regression calibration.

## 1 Introduction

Modern machine learning methods have witnessed great success on a wide range of tasks in the past decade for their high accuracy in dealing with various kinds of complicated data. Uncertainty quantification has played an important role in the interpretation and risk assessment of machine learning models for downstream tasks such as optimization and decision-making. People observe

---

[*]Equal contribution.

37th Conference on Neural Information Processing Systems (NeurIPS 2023).

that the output of deep learning models tends to be over over-confident [Guo et al., 2017, Amodei et al., 2016], which inspires many recent works on uncertainty calibration for classification problems [Kumar et al., 2019, Wenger et al., 2020, Luo et al., 2022]. Comparatively, there has been less systematic theoretical understanding of the uncertainty calibration for regression problems. For a regression problem, the output of a prediction model can be the prediction of the conditional mean (by minimizing mean squared error) or of the conditional median (by minimizing mean absolute error). But such a single-point prediction cannot characterize the uncertainty, which motivates the development of uncertainty calibration methods for regression problems. Some efforts have been made in order to estimate the entire distribution, which includes Bayesian ways [MacKay, 1992, Damianou and Lawrence, 2013], the frequentists' ways [Kendall and Gal, 2017, Lakshminarayanan et al., 2017, Cui et al., 2020, Zhao et al., 2020], and the nonparametric ways [Lei and Wasserman, 2014, Lei et al., 2018, Bilodeau et al., 2021, Song et al., 2019]. Another easier task is to predict the quantiles rather than the whole distribution. Existing ways mainly focus on completing the task of quantile prediction in one go [Pearce et al., 2018, Thiagarajan et al., 2020, Chung et al., 2021, Takeuchi et al., 2006, Stainwart and Christmann, 2011]. However, all those methods suffer from either statistical inconsistency under model misspecification or computational intractability during the training phase, or sometimes even both (see the detailed review in Appendix A).

In this paper, we suggest that previous ways of training a new quantile prediction model from scratch while discarding the pre-trained (mean) regression model may not be the best way for a quantile calibration objective, because the pre-trained model (though designed for a mean estimation objective) can be helpful for the quantile calibration. We propose a simple, natural, but theoretically non-trivial method that divides the whole quantile calibration task into two steps: (i) train a good regression model and (ii) estimate the conditional quantiles of its prediction error. Although a similar idea is applied in *split conformal prediction*[Papadopoulos et al., 2002, Vovk et al., 2005, Lei et al., 2018], the theoretical justification is still in a lack since the conformal prediction only requires an *average* calibration guarantee (see Definition 1) that can also be achieved without this first training step at all (where the detailed review is given in Appendix A). By a careful analysis of the *individual* calibration objective (see Definition 3), we capture the intuition behind the two-step procedure and formalize it in a mathematical guarantee. After a comprehensive numerical experiment on real-world datasets against existing benchmark algorithms, we suggest that one neither needs to estimate the whole distribution nor to train a new quantile model from scratch, while a pre-trained regression model and a split-and-subtract suffice, both theoretically and empirically. Our contribution can be summarized below:

First, we propose a simple algorithm that can estimate all percent conditional quantiles simultaneously. We provide the individual consistency of our algorithm and prove the minimax optimal convergence rate with respect to the mean squared error (Theorem 1 and 2). Our analysis is new and it largely relaxes the assumptions in the existing literature on kernel density estimation and order statistics. By showing the necessity of the Lipschitz assumption (Theorem 6), our result uses *minimal assumptions* to reach both *finite sample guarantee* and *individual calibration*, and our paper is the first to keep the latter two goals simultaneously up to our knowledge.

Second, we propose a two-step procedure of estimating "mean + quantile of error" rather than directly estimating the conditional quantiles, which enables a faster convergence rate both theoretically and empirically. Specifically, our convergence rate is of order $\tilde{O}(L^{\frac{2d}{d+2}} n^{-\frac{2}{d+2}})$, where $L$ is the Lipschitz constant of the conditional quantile with respect to the features, $n$ is the number of samples, and $d$ is the dimension of feature. Since the conditional mean function and the conditional quantile function are highly correlated, one can greatly reduce the Lipschitz constant by subtracting the mean from the quantile.

Moreover, we construct several simple examples to show the unexpected behavior of the existing calibration methods, suggesting that a population-level criterion such as sharpness or MMD can be misleading. As our analysis works as a positive result on individual calibration, we also provide a detailed discussion about the existing results on the impossibility of individual calibration, illustrating that their definitions of individual calibration are either impractical or too conservative.

## 2 Problem Setup

Consider the regression calibration problem for a given pre-trained model $\hat{f}$. We are given a dataset $\{(X_i, Y_i)\}_{i=1}^n \in \mathcal{X} \times \mathcal{Y}$ *independent* of the original data that trains $\hat{f}(x)$. Here $\mathcal{X} \in [0,1]^d \subset \mathbb{R}^d$ denotes the covariate/feature space and $\mathcal{Y} \subset \mathbb{R}$ denotes the response/label space. The i.i.d. samples $\{(X_i, Y_i)\}_{i=1}^n$ follow an unknown distribution $\mathcal{P}$ on $\mathcal{X} \times \mathcal{Y}$. We aim to characterize the uncertainty of the regression error $U_i := Y_i - \hat{f}(X_i)$ in a model agnostic manner, i.e., without any further restriction on the choice of the underlying prediction model $\hat{f}$.

Now we introduce several common notions of regression calibration considered in the literature. We first state all the definitions with respect to the error $U_i$ and then establish equivalence with respect to the original $Y_i$. Define the $\tau^{\text{-th}}$ quantile of a random variable $Z$ as $Q_\tau(Z) := \inf\{t : F(t) \geq \tau\}$ for any $\tau \in [0,1]$ where F is the c.d.f. of $Z$. Accordingly, the quantile of the conditional distribution $U|X = x$ is denoted by $Q_\tau(U|X = x)$. A quantile prediction/calibration model is denoted by $\hat{Q}_\tau(x) : \mathcal{X} \to \mathbb{R}$ which gives the $\tau$-quantile prediction of $U$ given $X = x$.

**Definition 1** (Marginal/Average Calibration). The model $\hat{Q}_\tau(x)$ is *marginally calibrated* if

$$\mathbb{P}(U \leq \hat{Q}_\tau(X)) = \tau$$

for any $\tau \in [0,1]$. Here the probability distribution is with respect to the joint distribution of $(U, X)$.

As noted by existing works, the marginal calibration requirement is too weak to achieve the goal of uncertainty quantification. A predictor that always predicts the marginal quantile of $U$ for any $X = x$, i.e., $\hat{Q}_\tau(x) \equiv Q_\tau(U)$, is always marginally calibrated, but such a model does not capture the heteroscedasticity of the output variable $Y$ or the error $U$ with respect to $X$.

**Definition 2** (Group Calibration). For some pre-specified partition $\mathcal{X} = \mathcal{X}_1 \cup \cdots \cup \mathcal{X}_K$, the model $\hat{Q}_\tau(x)$ is *group calibrated* if

$$\mathbb{P}(U \leq \hat{Q}_\tau(X)|X \in \mathcal{X}_k) = \tau$$

for any $\tau \in [0,1]$ and $k = 1, ..., K$.

Group calibration is a stronger notion of calibration than marginal calibrations. It is often considered in a related but different problem called *conformal prediction* [Vovk, 2012, Lei and Wasserman, 2014, Alaa et al., 2023] where the goal is to give a sharp covering set $\hat{C}(X)$ (or covering band if unimodality is assumed) such that $\mathbb{P}(Y \in \hat{C}(X)|X \in \mathcal{X}_k) \geq 1 - \alpha, \forall k$. Foygel Barber et al. [2021] consider the case where the guarantee is made for all $\mathbb{P}(X \in \mathcal{X}_k) \geq \delta$ for some $\delta > 0$.

**Definition 3** (Individual Calibration). The model $\hat{Q}_\tau(x)$ is *individually calibrated* if

$$\mathbb{P}(U \leq \hat{Q}_\tau(X)|X = x) = \tau$$

for any $\tau \in [0,1]$ and $x \in \mathcal{X}$. Here the probability distribution is with respect to the conditional distribution $U|X$.

In this paper, we consider the criteria of individual calibration, an even stronger notion of calibration than the previous two. It requires the calibration condition to hold with respect to any $x \in \mathcal{X}$ in a pointwise manner. The difference between the conformal prediction problem with group calibration and the regression calibration problem with individual calibration is that conformal prediction does not make any assumptions on the data distribution, while our lower bound result of Theorem 6 suggests that without any assumption, the convergence rate of the conditional quantile estimator can be arbitrarily slow. The conformal prediction literature considers a weaker notion of calibration than the individual calibration setting considered in this paper where some mild assumptions are made.

Finally, the following proposition says that any quantile prediction model on the error $U$ is equivalent to a corresponding quantile prediction on $Y$. The result is stated for the individual calibration, while a similar statement can be made for marginal calibration and group calibration as well.

**Proposition 1.** *For any predictor $\hat{Q}_\tau(x)$ on the $\tau^{\text{-th}}$ quantile of $U|X = x$, we have*

$$\mathbb{P}\left(Y \leq \hat{f}(X) + \hat{Q}_\tau(X)\big|X = x\right) = \mathbb{P}\left(U \leq \hat{Q}_\tau(X)\big|X = x\right).$$

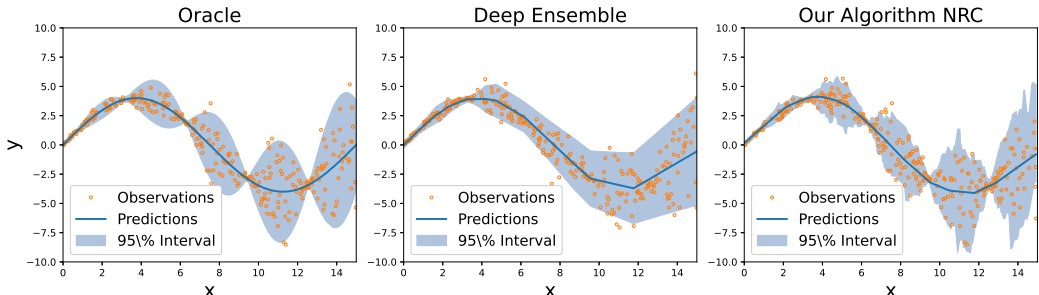

Figure 1: Synthetic data where the underlying distribution is obtained by a combination of sine functions. The solid lines denote predicted means, the shaded area denotes predicted intervals between $97.5\%$ and $2.5\%$ quantiles, and the yellow dots denote a subset of real observations. The leftmost plot gives the real mean as well as oracle quantile values, while the rest two plots are predictions from different calibration models. The middle plot is produced by a Deep Ensemble [Lakshminarayanan et al., 2017] of 5 HNNs trained with 40,000 samples, which is both a common benchmark and a building block for several regression calibration methods. The rightmost plot is produced by our proposed nonparametric calibration method – Algorithm 2 NRC of which the base regression model is an ordinary feed-forward regression network. The detailed setup is given in Appendix E.2.

## 2.1 Motivating examples

In addition to the above calibration objective, existing literature also considers sharpness (the average length of confidence intervals [Zhao et al., 2020, Chung et al., 2021]) and maximum mean discrepancy (MMD) principle [Cui et al., 2020]. To the best of our knowledge, all the existing works consider population-level objectives such as marginal calibration, sharpness, and MMD when training the calibration model, while we are the first work that directly aims for individual calibration. Here we use a simple example to illustrate that such population-level objectives may lead to undesirable calibration results, and in Appendix B, we elaborate with more examples on the inappropriateness of existing methods for regression calibration.

**Example 1.** Consider $X \sim \text{Unif}[0, 1]$ and the conditional distribution $Y|X = x \sim \text{Unif}[0, x]$. Then if one aims for $\tau = 90\%$, the outcome from sharpness maximization subject to marginal calibration is to $\hat{Q}_\tau(x) = x$ for $x \in [0, 0.9]$ and $\hat{Q}_\tau(x) = 0$ for $x \in (0.9, 1]$. Consequently, the quantile prediction has a $100\%$ coverage over $x \in [0, 0.9]$ but $0\%$ coverage over $x \in (0.9, 1]$.

Generally, the population-level criteria such as sharpness or MMD may serve as performance metrics to monitor the quality of a calibration outcome. However, the inclusion of such criteria in the objective function will encourage the calibration result to be over-conservative for the low variance region ($x \in [0, 0.9]$) while giving up the high variance ($x \in (0.9, 1]$), which is highly undesirable for risk-sensitive applications and/or fairness considerations.

Besides, individual calibration is sometimes also necessitated by the downstream application. For example, in some decision-making problems, regression calibration serves as a downstream task where loss is measured by the pinball loss; say, the newsvendor problem [Kuleshov et al., 2018, Ban and Rudin, 2019] in economics and operations research uses pinball loss to trade-off the backorder cost and the inventory cost.

**Proposition 2.** *The pinball loss with respect to $\tau$ is defined by*

$$l_\tau(u_1, u_2) := (1 - \tau)(u_1 - u_2) \cdot \mathbb{1}\{u_1 > u_2\} + \tau(u_2 - u_1) \cdot \mathbb{1}\{u_1 < u_2\}.$$

*We have*

$$Q_\tau(Y|X = x) = \hat{f}(x) + Q_\tau(U|X = x) \in \underset{y \in \mathbb{R}}{\arg\min}\, \mathbb{E}[l_\tau(y, Y)|X = x].$$

Proposition 2 serves as the original justification of pinball loss for quantile regression. More importantly, it says that a calibration method that is inconsistent with respect to individual calibration can be substantially suboptimal for downstream tasks such as the newsvendor problem.

# 3 Main Results

## 3.1 Generic nonparametric quantile estimator

In this subsection, we present and analyze a simple algorithm of a nonparametric quantile estimator, which can be of independent interest itself and will be used as a building block for the regression calibration. The algorithm takes the dataset $\{(X_i, U_i)\}_{i=1}^n$ as input and outputs a $\tau$-quantile estimation of the conditional distribution $U|X = x$ for any $\tau$ and $x$.

---

**Algorithm 1** Simple Nonparametric Quantile Estimator

---

**Input:** $\tau \in (0, 1)$, dataset $\{(X_i, U_i)\}_{i=1}^n$, kernel choice $\kappa_h(\cdot, \cdot)$
 1: Estimate the distribution of $U|X = x$ by (where $\delta_u$ is a point mass distribution at $u$)

$$\hat{\mathcal{P}}_{U|X=x} = \frac{\sum_{i=1}^n \kappa_h(x, X_i) \delta_{U_i}}{\sum_{i=1}^n \kappa_h(x, X_i)}.$$

 2: Output the conditional $\tau^{\text{-th}}$ quantile by minimizing the pinball loss on $\hat{\mathcal{P}}_{U|X=x}$

$$\hat{Q}_\tau^{\text{SNQ}}(x) = \arg\min_u \mathbb{E}_{U \sim \hat{\mathcal{P}}_{U|X=x}}[l_\tau(u, U)]. \tag{1}$$

---

The minimum of the optimization problem in Algorithm 1 is indeed achieved by the $\tau$-quantile of the empirical distribution $\hat{\mathcal{P}}_{U|X=x}$. In the following, we analyze the theoretical properties of the algorithm under the naive kernel

$$\kappa_h(x_1, x_2) = \mathbb{1}\{\|x_1 - x_2\| \le h\}.$$

where $h$ is the hyper-parameter for window size, and $\|x_1 - x_2\|$ denotes Euclidean distance.

**Assumption 1.** *We assume the following on the joint distribution $(U, X)$ and $\tau \in (0, 1)$ of interest:*

(a) *(Lipschitzness). The conditional quantile function is L-Lipschitz with respect to $x$,*

$$|Q_\tau(U|X = x_1) - Q_\tau(U|X = x_2)| \le L\|x_1 - x_2\|, \quad \forall x_1, x_2 \in \mathcal{X}.$$

(b) *(Boundedness). The quantile $Q_\tau(U|X = x)$ is bounded within $[-M, M]$ for all $x \in \mathcal{X}$.*

(c) *(Density). There exists a density function $p_x(u)$ for the conditional probability distribution $U|X = x$, and the density function is uniformly bounded away from zero in a neighborhood of the interested quantile. That is, there exist constants $\underline{p}$ and $\underline{r}$,*

$$p_x(u) \ge \underline{p}, \quad \forall x \in \mathcal{X} \text{ and } |u - Q_\tau(U|X = x)| \le \underline{r}.$$

In Assumption 1, part (a) and part (b) impose Lipschitzness and boundedness for the conditional quantile, respectively. Part (c) requires the existence of a density function and a lower bound for the density function around the quantile of interest. Its aim is to ensure a locally strong convexity for the expected risk $\mathbb{E}_{U \sim \mathcal{P}_{U|X=x}}[l_\tau(u, U)]$. Under Assumption 1, we establish consistency and convergence rate for Algorithm 1.

**Theorem 1.** *Under Assumption 1, Algorithm 1 is statistically consistent, i.e., for any $\epsilon > 0$,*

$$\lim_{n \to \infty} \mathbb{P}\left(\left|\hat{Q}_\tau^{SNQ}(X) - Q_\tau(U|X)\right| \ge \epsilon\right) = 0.$$

*Furthermore, by choosing $h = \Theta(L^{\frac{2}{d+2}} n^{-\frac{1}{d+2}})$, we have for sufficiently large $n \ge C$, when $L > 0$,*

$$\mathbb{E}\left[\left|\hat{Q}_\tau^{SNQ}(X) - Q_\tau(U|X)\right|^2\right] \le C' L^{\frac{2d}{d+2}} n^{-\frac{2}{d+2}},$$

*where $C$ and $C'$ depends polynomially on $\frac{1}{\underline{p}}$, $\frac{1}{\underline{r}}$, $M$, and $\log(n)$. In addition, whens $L = 0$, the right-hand-side becomes $\tilde{O}(\frac{1}{n})$.*

While Algorithm 1 seems to be a natural algorithm for nonparametric quantile regression, Theorem 1 provides the first convergence rate for such a nonparametric quantile estimator, to the best of our knowledge, in the literature of nonparametric regression; also, it is the first theoretical guarantee towards individual calibration for regression calibration. The standard analysis of nonparametric mean estimation [Györfi et al., 2002] cannot be applied here directly because the algorithm involves a local optimization (1). Even more challenging, the optimization problem (1) aims to optimize the quantile of the conditional distribution $U|X = x$ but the samples used in (1) are from distributions $U|X = x_i$, which causes a non-i.i.d. issue. To resolve these issues, we combine the idea of bias and variance decomposition in nonparametric analysis with a covering concentration argument for the local optimization problem. The detailed proof is deferred to Appendix H.

Theorem 1 also complements the existing results on finite-sample analysis for quantile estimators. One line of literature [Szorenyi et al., 2015, Altschuler et al., 2019] establishes the quantile convergence from the convergence of empirical processes, and this requires additional assumptions on the density function and does not permit the non-i.i.d. structure here. The quantile bandits problem also entails the convergence analysis of quantile estimators; for example, Zhang and Ong [2021] utilize the analysis of order statistics [Boucheron and Thomas, 2012], and the analysis inevitably requires a non-decreasing hazard rate for the underlying distribution. Other works [Bilodeau et al., 2021] that follow the kernel density estimation approach require even stronger conditions such as realizability. We refer to Appendix D for more detailed discussions. In comparison to these existing analyses, our analysis is new, and it only requires a minimal set of assumptions. The necessity of the assumptions can be justified from the following lower bound.

Theorem 2 rephrases the lower bound result in the nonparametric (mean) regression literature [Györfi et al., 2002] for the quantile estimation problem. It states that the convergence rate in Theorem 1 is minimax optimal (up to poly-logarithmic terms). It is easy to verify that the class $\mathcal{P}^L$ satisfies Assumption 1. In this light, Theorem 1 and Theorem 2 establish the criticalness of Assumption 1. Furthermore, in Theorem 6 in Appendix C, we establish that the convergence rate of any estimator can be arbitrarily slow without any Lipschitzness assumption on the conditional statistics.

**Theorem 2.** *Let $\mathcal{P}^L$ be the class of distributions of $(X, U)$ such that $X \sim \mathrm{Uniform}[0, 1]^d$, $U = \mu(X) + N$, and $\mu(x)$ is L-Lipschitz where $N$ is an independent standard Gaussian random variable. For any algorithm, there exists a distribution in the class $\mathcal{P}^L$ such that the convergence rate of the algorithm is $\Omega(L^{\frac{2d}{d+2}} n^{-\frac{2}{d+2}})$.*

Our result here provides a guarantee for the estimation of $Q_\tau(U|X)$ and thus a positive result for individual calibration with respect to a specific $\tau$. We note that the result does not contradict the negative results on individual calibration [Zhao et al., 2020, Lei and Wasserman, 2014]. Zhao et al. [2020] measure the quality of individual calibration via the closeness of the distribution $\hat{F}_X(Y)$ to the uniform distribution. In fact, such a measurement is only meaningful for a continuous distribution of $Y|X$, but they prove the impossibility result based on a discrete distribution of $Y|X$. So, their negative result on individual calibration does not exclude the possibility of individual calibration for a continuous $Y|X$. Lei and Wasserman [2014] require an almost surely guarantee based on finite observations and prove an impossibility result. This is a very strong criterion; our results in Theorem 1 are established for two weaker but more practical settings: either the strong consistency in the asymptotic case as $n \to \infty$ or a mean squared error guarantee under finite-sample. We defer more discussions to Appendix C.

### 3.2 Regression calibration

Now we return to regression calibration and build a calibrated model from scratch. Specifically, Algorithm 2 splits the data into two parts; it uses the first part to train a prediction model and the second part to calibrate the model with Algorithm 1. The first part can be skipped if there is already a well-trained model $\hat{f}$ where it becomes a recalibration problem.

The theoretical property of Algorithm 2 can be established by combining the results from Section 3.1 with Proposition 1. In Algorithm 2, the quantile calibration allows full flexibility in choosing the regression model $\hat{f}$ and does not require an associated uncertainty model to proceed. The motivation for calibrating the error $U_i$ instead of the original $Y_i$ is two-fold: First, if one treats $Y_i$ itself as $U_i$ and applies Algorithm 1, then it essentially restricts the prediction model to be a nonparametric one. Second, the reduction from $Y_i$ to $U_i$ gives a better smoothness of the conditional distribution (a

---

**Algorithm 2** Nonparametric Regression Calibration

---

**Input:** Dataset $\{(X_i, Y_i)\}_{i=1}^n$, kernel choice $\kappa_h(\cdot, \cdot)$, $\tau$
  1: Split the dataset into half: $\mathcal{D}_1 = \{(X_i, Y_i)\}_{i=1}^{n_1}$, $\mathcal{D}_2 = \{(X_i, Y_i)\}_{i=n_1+1}^n$.
  2: Use $\mathcal{D}_1$ to train a regression model $\hat{f}$.
  3: Calculate the estimation error of $\hat{f}$ on $\mathcal{D}_2$: $U_i = Y_i - \hat{f}(X_i)$, $i = n_1 + 1, \cdots, n$.
  4: Run Algorithm 1 on the data $\{(X_i, U_i)\}_{i=n_1+1}^n$ and obtain $\hat{Q}_\tau^{\text{SNQ}}(\cdot)$.
  5: Return $\hat{f}(\cdot) + \hat{Q}_\tau^{\text{SNQ}}(\cdot)$.

---

smaller Lipschitz constant in Assumption 1 (a)). And thus it will give a faster convergence rate. We remark that Algorithm 2 induces an independence between the training of the prediction model and that of the calibration model. The design is intuitive and important because otherwise, it may result in predicting smaller confidence intervals both theoretically and empirically.

### 3.3 Implications from the nonparametric perspective

The nonparametric approach gives a positive result in terms of the possibility of individual calibration, but it pays a price with respect to the dimensionality $d$. The following theorem states that such a dimensionality problem can be addressed under conditional independence. And more generally, it provides a guideline on which variables one should use to perform calibration tasks.

**Theorem 3.** *Suppose $U = Y - \hat{f}(X)$. Suppose $Z = m(X) \in \mathcal{Z}$, where $m$ is some measurable function, and $\mathcal{Z} \subset \mathcal{X}$ is a $d_0$-dimensional subspace of $\mathcal{X}$. If $X$ and $U$ are mutually independent conditioned on $Z$ (i.e. $X \to Z \to U$ is a Markov chain), then it is lossless to perform calibration using only $(U_i, Z_i)$'s. In particular, applying Algorithm 1 on $(U_i, Z_i)$'s will yield the same consistency but with a faster convergence rate of $O(n^{-\frac{2}{d_0+2}})$.*

Theoretically, one can identify such $Z$ by independence test for each component of $X$ and $Y - \hat{f}(X)$ on the validation set. Some other practical methods such as random projection and correlation screening selection are shown in Section 4. When the conditional independence in Theorem 3 does not hold, it is still not the end of the world if one calibrates $(U_i, Z_i)$. The following theorem tells that conditional calibration with respect to $Z$, as an intermediate between marginal calibration and individual calibration, can be achieved.

**Theorem 4.** *Suppose $U = Y - \hat{f}(X)$ and $Z$ is a sub-feature of $X$. Without loss of generality, we assume that $Z = \Pi_{d_0}(X)$, where $\Pi_{d_0}$ projects from $\mathcal{X}$ onto a $d_0$-dimensional subspace $\mathcal{Z}$. Then*

$$\mathbb{P}(U \leq Q_\tau(U|Z) \mid Z) = \tau,$$

*which implies that $\mathbb{P}(Y \leq \hat{f}(X) + Q_\tau(U|Z) \mid Z) = \tau$.*

A notable implication from the above theorem is that if we want to calibrate the model against certain sub-features such as age, gender, geo-location, etc., we can summarize these features in $Z$ and use Algorithm 1 with respect to $(U_i, Z_i)$.

However, if we return to the original pinball loss, the following theorem inherits from Proposition 2 and states that the inclusion of more variables will always be helpful in terms of improving the pinball loss. It highlights a trade-off between the representation ability of the quantile calibration model (inclusion of more variables)and its generalization ability (slowing down the convergence rate with higher dimensionality).

**Theorem 5.** *Suppose $U = Y - \hat{f}(X)$. Suppose $Z^{(d_1)}$ and $Z^{(d_2)}$ are the first $d_1$ and $d_2$ components of $X$. Suppose $Q_\tau(F_{Z^{(d_i)}})$ is the true $\tau^{\text{-th}}$ quantile of $U$ conditioned on $Z^{(d_i)}$, $i = 1, 2$. Suppose $Q_\tau(F_{margin})$ is the true $\tau^{\text{-th}}$ marginal quantile of $U$. If $0 < d_1 < d_2 < d$, then*

$$\mathbb{E}[l_\tau(\hat{f}(X) + Q_\tau(U), Y)] \geq \mathbb{E}[l_\tau(\hat{f}(X) + Q_\tau(U|Z^{(d_1)}), Y)]$$
$$\geq \mathbb{E}[l_\tau(\hat{f}(X) + Q_\tau(U|Z^{(d_2)}), Y)] \geq \mathbb{E}[l_\tau(\hat{f}(X) + Q_\tau(U|X), Y)].$$

# 4 Numerical Experiments

In this section, we evaluate our methods against a series of benchmarks. We first introduce the evaluation metrics that are widely considered in the literature of uncertainty calibration and conformal prediction, including Mean Absolute Calibration Error (MACE) which calculates the average absolute error for quantile predictions from 0.01 to 0.99; Adversarial Group Calibration Error (AGCE) which finds the sub-group of the test data with the largest MACE; Check Score which shows the empirical pinball loss; Length which measures the average length for the constructed 0.05-0.95 intervals; and Coverage which reflects the empirical coverage rate for those 0.05-0.95 intervals. Formal definitions of these calibration measurements are provided in Appendix E.1.

The benchmark methods are listed as follows (the details can be found in Appendix E.4). For the Gaussian-based methods, we implement the vanilla Heteroscedastic Neural Network (HNN) [Kendall and Gal, 2017], Deep Ensembles (DeepEnsemble) [Lakshminarayanan et al., 2017], and Maximum Mean Discrepancy method (MMD) [Cui et al., 2020]. We also implement MC-Dropout (MCDrop) [Gal and Ghahramani, 2016] and Deep Gaussian Process model (DGP) [Damianou and Lawrence, 2013]. For models that do not rely on Gaussian assumption, we implement the combined calibration loss (CCL) introduced in Sec. 3.2 of [Chung et al., 2021]. We also implement post-hoc calibration methods, such as isotonic regression (ISR) suggested by [Kuleshov et al., 2018] with a pre-trained HNN model. Another post-hoc method named Model Agnostic Quantile Regression (MAQR) method in Sec. 3.1 of Chung et al. [2021] is also implemented. For conformal prediction algorithms, we implement Conformalized Quantile Regression (CQR) [Romano et al., 2019] and Orthogonal Quantile Regression (OQR) [Feldman et al., 2021].

As our proposed nonparametric regression calibration (NRC) algorithm is agnostic with respect to the underlying prediction model, we implement a feed-forward neural network and a random forest model as the prediction model for NRC. Also, we apply the dimension reduction idea (called NRC-DR) including random projection and covariate selection where all technical details are given in Appendix E.3 and E.5. All codes are available at `https://github.com/ZhongzeCai/NRC`.

In the following subsections, we summarize our results on UCI datasets, time series, and high-dimensional datasets. Additional experiments on covariate shift are deferred to E.7.

## 4.1 UCI dataset experiments

We evaluate our NRC algorithms on the standard 8 UCI datasets Dua and Graff [2017]. Part of the experiment results on representative benchmark models is given in Table 1, and the full results are presented in Appendix E.5 due to space limit. Additional details on dataset description, hyperparameter selection, and early stopping criteria are given in Appendix E.5.

The result supports the competitiveness of our algorithm. In terms of MACE, our methods achieve the minimum loss on 6 out of 8 datasets, with the non-optimal results close to the best performance seen on other benchmarks. The rest metrics also witness our algorithms outperforming the benchmark models, with a superior result on 6 datasets in terms of AGCE, and 4 in terms of CheckScore. Note that even for the rest 4 datasets where our NRC algorithm does not achieve the best CheckScore (the empirical pinball loss), our performance is still competitive compared to the best score, while for some specific datasets (such as Energy and Yacht) we achieve a significant improvement over the CheckScore, which verifies the potential of our methods for the related downstream tasks. As for the Length metric (which evaluates the average interval length of a symmetric $90\%$ confidence interval), it is not easy to trade-off between the sharpness requirement and the coverage rate due to the fact that one can trivially sacrifice one for the other, so we omit the direct comparison. However, we can still observe that our NRC algorithm behaves reasonably well, with a stable coverage rate performance and a relatively sharp interval length amongst the rest benchmarks.

## 4.2 Bike-sharing time series dataset

This experiment explores the potential of our algorithm to quantify the uncertainty of the time series objective, where independent variables are aggregated in a sliding window fashion from the past few days. The Bike-Sharing dataset provided in Fanaee-T and Gama [2014] is used to visually evaluate our proposed algorithm. Due to the limitations in the representation ability of the feed-forward structure, we instead deploy LSTM networks for the time series data. We design LSTM-HNN,

Table 1: Experiments on the standard UCI datasets. Each experiment (for a combination of dataset and method) is repeated 5 times in order to estimate the range of value fluctuation of each metric. On each dataset, the minimum loss for each metric is marked in bold and red, while we omit the comparisons on the Length and the Coverage due to the difficulty of trading off between sharper intervals and better coverage intervals. Full experiment details are deferred to Appendix E.

| Dataset | Metric | MCDrop | DeepEnsemble | ISR | MAQR | CQR | NRC | NRC-DR |
|---|---|---|---|---|---|---|---|---|
| Boston | MACE | 0.14±0.03 | 0.06±0.04 | 0.057±0.03 | 0.051±0.02 | 0.049±0.02 | **0.048±0.02** | 0.055±0.02 |
| | AGCE | 0.22±0.04 | 0.27±0.07 | 0.27±0.08 | 0.25±0.06 | 0.24±0.06 | 0.19±0.05 | **0.19±0.05** |
| | CheckScore | 1.9±0.2 | 1±0.3 | 0.95±0.2 | 1.1±0.2 | **0.72±0.1** | 1.1±0.2 | 1.1±0.2 |
| | Length(Coverage) | 8.6±1(95%) | 8.5±1(90%) | 7.9±1(88%) | 9.1±1(88%) | 7.1±0.7(82%) | 10±1(94%) | 9.7±1(91%) |
| Concrete | MACE | 0.078±0.02 | 0.053±0.03 | 0.044±0.02 | 0.049±0.03 | 0.059±0.03 | 0.047±0.02 | **0.038±0.03** |
| | AGCE | 0.2±0.04 | **0.18±0.03** | 0.19±0.05 | 0.18±0.06 | 0.23±0.05 | 0.18±0.05 | 0.19±0.04 |
| | CheckScore | 3.6±0.2 | 1.9±0.3 | 2.3±0.6 | 2±0.2 | 1.72±0.6 | **1.4±0.2** | 2±0.4 |
| | Length(Coverage) | 25±4(95%) | 19±1(93%) | 20±0.8(89%) | 17±2(79%) | 20±0.7(89%) | 21±0.8(90%) | 19±0.9(92%) |
| Energy | MACE | 0.16±0.02 | 0.065±0.04 | 0.055±0.03 | 0.063±0.02 | 0.075±0.03 | **0.037±0.008** | 0.04±0.01 |
| | AGCE | 0.21±0.01 | **0.19±0.04** | 0.21±0.07 | 0.21±0.04 | 0.18±0.008 | 0.22±0.06 | 0.24±0.08 |
| | CheckScore | 1.4±0.3 | 0.64±0.09 | 0.6±0.1 | 0.6±0.2 | **0.14±0.03** | 0.18±0.03 | 0.69±0.2 |
| | Length(Coverage) | 9±4(93%) | 7.7±0.6(94%) | 7.1±2(95%) | 6.2±3(84%) | 1.8±0.3(95%) | 4.5±1(88%) | 1.5±0.1(89%) |
| Kin8nm | MACE | 0.034±0.01 | 0.077±0.05 | 0.013±0.005 | 0.032±0.02 | 0.041±0.02 | **0.013±0.004** | 0.017±0.007 |
| | AGCE | 0.075±0.01 | 0.12±0.06 | 0.072±0.02 | 0.072±0.02 | 0.075±0.02 | 0.067±0.02 | **0.056±0.02** |
| | CheckScore | 0.045±0.002 | **0.027±0.002** | 0.031±0.001 | 0.029±0.001 | 0.025±0.0009 | 0.032±0.002 | 0.032±0.002 |
| | Length(Coverage) | 0.5±0.02(95%) | 0.29±0.006(93%) | 0.28±0.007(90%) | 0.27±0.01(87%) | 0.26±0.005(89%) | 0.25±0.003(90%) | 0.52±0.007(90%) |
| Naval | MACE | 0.1±0.03 | 0.12±0.05 | **0.011±0.005** | 0.043±0.02 | 0.077±0.07 | 0.012±0.005 | 0.017±0.007 |
| | AGCE | 0.15±0.05 | 0.14±0.06 | 0.054±0.009 | 0.081±0.01 | 0.098±0.05 | **0.052±0.008** | 0.052±0.008 |
| | CheckScore | 0.0018±0.0006 | **0.00064±0.0002** | 0.00079±0.0003 | 0.0021±0.0007 | 0.0011±0.0009 | 0.00088±0.0001 | 0.002±0.0007 |
| | Length(Coverage) | 0.02±0.002(94%) | 0.015±0.0003(100%) | 0.0078±0.0008(90%) | 0.017±0.003(84%) | 0.015±0.002(98%) | 0.0097±0.002(91%) | 0.0088±0.0006(91%) |
| Power | MACE | 0.29±0.01 | 0.045±0.03 | 0.011±0.006 | 0.029±0.02 | 0.049±0.005 | **0.0086±0.003** | 0.0086±0.003 |
| | AGCE | 0.3±0.01 | 0.075±0.03 | 0.065±0.02 | 0.06±0.01 | 0.071±0.02 | 0.057±0.007 | **0.053±0.008** |
| | CheckScore | 24±3 | 1.3±0.1 | 1.3±0.1 | 1.2±0.03 | 1.1±0.03 | **1±0.03** | 1.2±0.05 |
| | Length(Coverage) | 20±3(97%) | 16±0.5(97%) | 13±0.8(90%) | 12±0.8(88%) | 13±0.7(93%) | 13±0.3(91%) | 11±0.1(90%) |
| Wine | MACE | 0.036±0.02 | 0.049±0.03 | 0.016±0.009 | 0.029±0.02 | 0.056±0.007 | **0.016±0.01** | 0.017±0.006 |
| | AGCE | 0.1±0.03 | 0.12±0.05 | 0.091±0.02 | 0.094±0.03 | 0.1±0.06 | 0.075±0.03 | **0.075±0.02** |
| | CheckScore | 0.2±0.01 | 0.2±0.009 | 0.2±0.01 | 0.2±0.02 | 0.21±0.02 | **0.19±0.01** | 0.21±0.01 |
| | Length(Coverage) | 2.4±0.09(90%) | 2.2±0.02(88%) | 2.2±0.03(87%) | 2.3±0.1(84%) | 2.3±0.04(89%) | 2.3±0.04(90%) | 2.2±0.02(90%) |
| Yacht | MACE | 0.11±0.03 | **0.049±0.02** | 0.06±0.03 | 0.088±0.04 | 0.073±0.05 | 0.096±0.05 | 0.087±0.04 |
| | AGCE | **0.29±0.08** | 0.35±0.06 | 0.34±0.09 | 0.31±0.07 | 0.34±0.04 | 0.34±0.07 | 0.31±0.04 |
| | CheckScore | 1.9±0.6 | 1.1±0.4 | 1.5±0.6 | 0.32±0.09 | 0.19±0.07 | **0.18±0.03** | 0.4±0.2 |
| | Length(Coverage) | 18±5(94%) | 10±1(91%) | 10±0.9(88%) | 3.3±1(83%) | 1.7±0.4(72%) | 5.3±1(93%) | 3.2±1(91%) |

which is an ordinary LSTM network followed by a fully connected layer with 2 output units; and LSTM-NRC, whose regression model shares the same LSTM setting as LSTM-HNN except for the ending layer having only one output unit (without variance/uncertainty unit). For more details, see Appendix E.6.

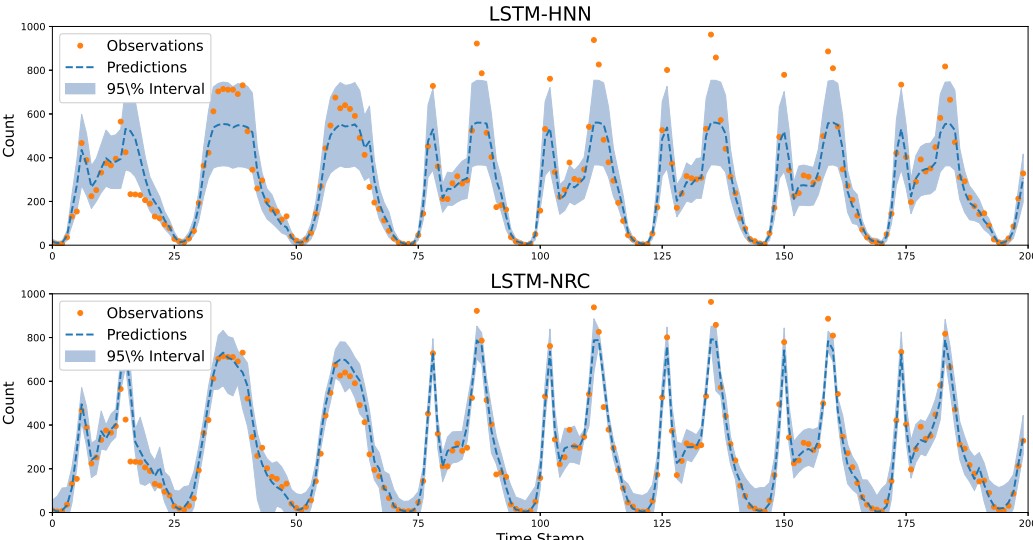

Figure 2: Experiments on the Bike-Sharing dataset. The demonstrated time stamps are consecutively taken from the test set, thus temporally separated from the training set. The top figure gives the predicted 95% confidence intervals of an LSTM-HNN model, and the bottom figure is produced from a random projection variant of our NRC method (LSTM-NRC). The target variable shows strong periodicity, which both two models capture. Nevertheless, our NRC method delivers sharper confidence intervals and is better adapted to the underlying distribution shift. In particular, our method also has a better coverage rate on the extreme values at time stamps between 100 and 200.

Figure 2 visually compares the two models on the $95\%$ confidence interval prediction task on the test set (see Appendix E.6 for how the test set is split out). A first observation is that our LSTM-NRC is producing sharper and better calibrated $95\%$ intervals. Also, in spite of the satisfying coverage rate of both the methods on time range 0 to 100 (time stamp -1 is exactly the terminating point of the training set, thus time range 0 to 100 roughly denotes the near future of the historical events), on time range 100 to 200 the behaviors of the two models diverge. After an evolution of 100 time stamps the target variable (rental count per hour) is under a shift in distribution: the peak values get higher than historical records would foresee. Confidence intervals produced by LSTM-HNN fail to cover these "extremer" peaks, while our LSTM-NRC method could easily adapt to it, taking advantage of a high-quality regression model. Note that both LSTM-HNN and LSTM-NRC share the same regression architecture, we argue that the additional variance unit of LSTM-HNN may hurt the mean prediction performance of the model.

## 4.3 Higher-dimensional datasets

One major drawback of the non-parametric type methods is its unsatisfying dependence on the dimension of the data. Our paper attacks such an obstacle from two aspects simultaneously: on the one hand, we subtract a regression model from the original data to gain a smaller Lipschitz constant $L$; on the other hand, we apply several dimensional reduction methods to reduce the data dimension manually. In this paper, we mainly implement three of them: random projection, covariate selection, and second-to-last layer feature extraction, of which the details can be found in Appendix E.3. We call the variants of our NRC algorithm NRC-RP, NRC-Cov, and NRC-Embed, respectively. We examine them against several high-dimensional datasets: the medical expenditure panel survey (MEPS) datasets [panel 19, 2017, panel 20, 2017, panel 21, 2017], as suggested in [Romano et al., 2019]. The datasets' details are also deferred to Appendix E.3.

| Dataset | Metric | DeepEnsemble | ISR | CQR | OQR | NRC-RP | NRC-Embed |
|---------|--------|--------------|-----|-----|-----|--------|-----------|
| meps_19 | MACE | 0.26±0.04 | 0.12±0.05 | 0.05±0.01 | 0.055±0.02 | 0.024±0.01 | **0.0087±0.006** |
|  | AGCE | 0.26±0.04 | 0.14±0.04 | 0.079±0.01 | 0.091±0.03 | 0.062±0.02 | **0.04±0.02** |
|  | CheckScore | 25±7 | 10±6 | **2.4±0.2** | 2.5±0.2 | 3.2±0.4 | 2.4±0.7 |
|  | Length(Coverage) | 230±140(99%) | 50±6(93%) | 28±2(89%) | 25±2(85%) | 35±8(90%) | 24±0.5(90%) |
| meps_20 | MACE | 0.25±0.02 | 0.16± 0.08 | 0.046±0.01 | 0.057±0.024 | **0.0052±0.004** | 0.0058±0.005 |
|  | AGCE | 0.25±0.02 | 0.18±0.08 | 0.061±0.01 | 0.08±0.03 | **0.033±0.01** | 0.041±0.007 |
|  | CheckScore | 23±8 | 7.4±1 | 2.5±0.1 | 2.5±0.1 | 3±0.4 | **2.4±0.08** |
|  | Length(Coverage) | 190±50(99%) | 69±38(94%) | 28±2(89%) | 27±4(85%) | 32±5(90%) | 27±3(91%) |
| meps_21 | MACE | 0.25±0.02 | 0.13±0.1 | 0.05±0.02 | **0.0087±0.004** | 0.012±0.002 | 0.0088±0.005 |
|  | AGCE | 0.26±0.02 | 0.15±0.1 | 0.063±0.008 | 0.075±0.02 | 0.045±0.02 | **0.041±0.01** |
|  | CheckScore | 36±20 | 23±10 | **2.5±0.3** | 2.6±0.3 | 3.5±0.7 | 2.7±1 |
|  | Length(Coverage) | 160±40(99%) | 43±20(93%) | 27±3(86%) | 29±3(89%) | 39±3(90%) | 24±2(91%) |

Table 2: Experiments on MEPS Datasets

In Table 2 we summarize the dimension reduction variants of our NRC algorithm against several benchmarks. The result sees a substantial advantage of our method over the selected benchmark methods, while most of the unshown benchmarks either are too computationally expensive to run or do not deliver competitive results on the high-dimensional dataset.

## 5 Conclusion

In this paper, we propose simple nonparametric methods for regression calibration, and through its lens, we aim to gain a better theoretical understanding of the problem. While numerical experiments show the advantage, we do not argue for a universal superiority of our methods. Rather, we believe the contribution of our work is first to give a positive result for individual calibration and an awareness of its importance, and second to show the promise of returning to simple statistical estimation approaches than designing further complicated loss functions without theoretical understanding. Future research includes (i) How to incorporate the uncertainty output from a prediction model with our nonparametric methods; (ii) While we do not observe the curse of dimensionality for our methods numerically and we explain it by the reduction of Lipschitzness constant, it deserves more investigation on how to adapt the method for high-dimensional data.

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

# A  Related Literature

## A.1  Estimating the whole distribution

The most straightforward thought is to estimate the whole distribution which accomplishes the task of mean regression and uncertainty quantification simultaneously. Generally, there are two ways to fit a distribution: either fitting a parametric model or turning to nonparametric methods. Representative parametric methods are Bayesian Neural Networks [MacKay, 1992] and Deep Gaussian Process (DGP) [Damianou and Lawrence, 2013], of which the latter assumes the data is generated according to a Gaussian process of which the parameters are further generated by another Gaussian process. The major drawbacks of these Bayesian-type methods are their high computation cost during the training phase and statistical inconsistency under model misspecification. To resolve the computational issue, [Gal and Ghahramani, 2016] proposes a simple MC-Dropout method that captures the model uncertainty without changing the network structure. Despite the view of Bayesian, the task of distribution estimation also falls into the range of frequentists: by assuming the underlying distribution family (say, Gaussian), one can fulfill the task by estimating the moments or maximizing the likelihood. For example, the Heteroscedastic Neural Network (HNN) [Kendall and Gal, 2017] gives an estimation of both mean and variance at the final layer by assuming the Gaussian distribution. An ensemble method called Deep Ensemble [Lakshminarayanan et al., 2017] is constructed on the basis of HNN. [Cui et al., 2020] designs an algorithm that finds the best Gaussian distribution to minimize the empirical Maximum Mean Discrepancy (MMD) from the observed data so as to ensure the marginal calibration property of quantiles. [Zhao et al., 2020] raises a randomized forecasting principle that minimizes the randomized calibration error with a negative log-likelihood regularization term to ensure the so-called sharpness. Although they claim the construction of (randomized) individual calibration (named PAIC) from the existence of mPAIC predictors, they do not provide any non-trivial mPAIC predictor except for the one that permanently predicts the uniform distribution.

The other approach to estimating the entire distribution is the nonparametric way without assuming the underlying distribution. For example, Kernel Density Estimation (KDE) utilizes kernel methods to estimate the conditional probability density function. Although the method enjoys certain theoretical consistency guarantees [Lei and Wasserman, 2014, Lei et al., 2018, Bilodeau et al., 2021], it is often computationally expensive and not sample-efficient in high dimensional cases, which makes it impractical for uncertainty quantification. A similar idea is applied in [Song et al., 2019] via a post-hoc way. The method is to improve on some given distribution estimation via the Gaussian process and Beta link function. However, unlike the KDE literature, the method in [Song et al., 2019] lacks a theoretical guarantee of its consistency.

## A.2  Estimating quantiles

Due to the excessive and sometimes unnecessary difficulties in estimating the distribution, some researchers suggest estimating the quantiles only, since a satisfying quantile estimation suffices in many downstream tasks. The quantile prediction is now regarded as a new task, where one natural idea is to combine the average calibration error on quantiles with a sharpness regularization term. [Pearce et al., 2018] appends a calibration error (with respect to coverage and sharpness) to the loss functions to obtain the calibrated prediction intervals at a pre-specified confidence level. [Thiagarajan et al., 2020] induces an extra interval predictor with a similar combined loss. But their methods require the pre-determined confidence or quantile level and the whole model needs to be retrained if one wants another level of prediction. Such methods in [Pearce et al., 2018, Thiagarajan et al., 2020] are only evaluated empirically without theoretical guarantee for even the marginal distribution. [Chung et al., 2021] proposes an average calibration loss combined with a similar sharpness regularization term, which achieves an average (but not individually, as we will discuss later) calibration. Along with these models that aim to give a quantile prediction out of scratch, [Kuleshov et al., 2018] develops a post-hoc way of calibrating (on average) a pre-trained quantile prediction model via isotonic regression methods. The aim is to choose a specific level of quantile prediction that reaches the desired marginal calibration of which the consistency relies solely on the pre-trained model in contrast to our model that applies a model-agnostic post-hoc calibration.

Another line of research is to use Kernelized Support Vector Machines (KSVM) to predict the conditional quantiles in one step, which fixes the inconsistency issue [Takeuchi et al., 2006, Stainwart and Christmann, 2011]. Essentially, methods from both papers search for a quantile prediction

function over the Reproducing Kernel Hilbert Space (RKHS), say $\mathcal{H}$. Theoretically, Takeuchi et al. [2006] derive a performance guarantee for the conditional quantile estimator based on the assumption of finite $\mathcal{H}$-norm, which is in parallel with our Lipschitz class (Assumption 1 (a)). The Lipschitz function class and the bounded $\mathcal{H}$-norm function class overlap, but neither one is contained in the other. Stainwart and Christmann [2011] analyze the same algorithm but adopt a different approach. Instead of imposing a bounded $\mathcal{H}$-norm, they impose a decay rate for the eigenvalues of the kernel integral operator, alongside some other assumptions on the underlying distribution. Those works focus on the RKHS space while we focus on the Lipschitz space. For the Lipschitz function class, to the best of our knowledge, our work is the first result to provide an individual guarantee. From the practical point of view, both papers rely on solving a kernelized learning problem that cannot scale well with the sample size or dimension, while our algorithm is simple to implement and of low computational cost.

### A.3 Conformal prediction

A very closely related field is conformal prediction, where the goal is to give a coverage set that contains the true label with a certain probability. Theoretically, the goal of conformal prediction is no harder than our quantile prediction task since correct quantiles induce desired coverage sets, while empirically different measurements are considered: the conformal prediction cares about giving the sharpest covering set rather than specifying which quantiles are estimated. For the regression problem, there is an "impossible triangle" summarized in a concurrent work [Gibbs et al., 2023]: (i) a conditional/individual coverage (as our Definition 3) (ii) no assumptions on the underlying distribution (iii) a finite-sample guarantee and asymptotic consistency. Such a triangle is shown by Vovk [2012], Foygel Barber et al. [2021] to be impossible to achieve simultaneously. Some works in the conformal prediction literature aim to reconcile the impossibility: Romano et al. [2019] consider the marginal/average case (as defined in Definition 1), which greatly relaxes (i). Gibbs et al. [2023] violates (i) in a milder but different way, focusing on a midway between marginal coverage and conditional coverage defined by the linear function class.

The most related paper to ours may be Feldman et al. [2021]: both theirs and ours attempt to address the minimum cost of keeping the requirement (i). Before diving into the detailed comparison, we summarize the main difference first: Feldman et al. [2021] relaxes the remaining two, while our paper keeps (iii) and relaxes (ii) in a much milder way. Feldman et al. [2021] achieves a theoretical guarantee that the true conditional quantile is the minimizer of the regularized population loss under a realizability assumption (that is, the true conditional quantiles lie in the function class considered) and an assumption that the conditional distribution is continuous with respect to features. However, its theoretical results have several limitations. First, it achieves only asymptotic consistency compared to our finite-sample guarantee in Theorem 1, which means that Feldman et al. [2021] violates requirement (iii). Besides, it only achieves a necessary condition that the true conditional quantile is the minimizer of the population loss but not vice versa. Second, Feldman et al. [2021] violates (ii) to a much deeper content than ours. Note that the key assumptions made in Feldman et al. [2021] are the zero-approximation-error assumption and the continuity assumption. The former means that there should be no model misspecification at all, which is not a weak one in the statistical learning theory and limits the guarantee from a desirable "distribution-free" property. On the contrary, our algorithm can have a finite-sample guarantee with only three very general assumptions summarized in Assumption 1. Furthermore, our Lipschitz assumption is made only to achieve a finite-sample guarantee (of which the necessity is supported by Theorem 6), while it can still be relaxed to a weaker one than Feldman et al. [2021] to reach the same asymptotic result. To see that, our result only requires the conditional quantile to be continuous, while Feldman et al. [2021] requires continuity for the whole conditional distribution. Third, the regularization term in Feldman et al. [2021] is related to a zero-one indicator, which is discontinuous for gradient descent. The authors twist their algorithm by replacing it with a smooth approximation to overcome the issue without any formal guarantee on the twisted one. In contrast, our theory accompanies the exact same algorithm that we propose.

Our paper can also serve as a starting point for understanding the empirical success of the "split" procedure that is common among the conformal prediction algorithms. As summarized before, most conformal prediction methods theoretically abandon requirement (i) with only a marginal coverage guarantee. But a tricky point is that if one only needs the marginal coverage, it suffices to merely use the empirical quantile without the splitting step at all. Why does the learner bother to "split"? Empirically, the splitting procedure helps to obtain a more homogeneous coverage across the whole

feature space, while our paper is the first theoretical justification up to our knowledge: to reduce the Lipschitz constant $L$ and smooth the target. Such an intuition is justified by Theorem 1 and Theorem 2 from both the upper and the lower bounds.

## B  Failure of Existing Algorithms in Individual Calibration

In this subsection, we will give a brief discussion of the existing methods in regression calibration literature. We give a close inspection of the existing calibration algorithms, proving that they cannot achieve a desirable result (or even avoid naive and meaningless ones).

Some methods assume that the underlying distribution is Gaussian [Cui et al., 2020, Zhao et al., 2020]. The most straightforward drawback one can think of is that these Gaussian-based methods are not consistent once the Gaussian assumption is violated. However, we will prove with some counter-examples that they cannot prevent the naive or meaningless prediction even if the underlying model is Gaussian.

Cui et al. [2020] use a two-step procedure to estimate the conditional distribution, where at each step the underlying model is a neural network predicting the mean $\mu(x)$ and variance $\sigma^2(x)$. The first step is simply fitting the observed $\{(X_i, Y_i)\}_{i=1}^n$ into the model, getting an initial guess of $\mu$ and $\sigma$. The second step utilizes the maximum mean discrepancy (MMD) principle. To be more specific, the second step iterates the following steps until convergence: first generate $N$ new samples $\{\hat{Y}_i\}_{i=1}^N$ from the estimated Gaussian model $\mathcal{N}(\hat{\mu}(X_i), \hat{\sigma}^2(X_i))$, then update the model estimation $(\hat{\mu}, \hat{\sigma}^2$ to minimize the empirical MMD $\|\frac{1}{N}\sum_{i=1}^N \phi(Y_i) - \frac{1}{N}\sum_{i=1}^N \phi(\hat{Y}_i)\|$. But the problem is: the corresponding relation between each $\{(X_i, Y_i)\}$ pair is wiped out from the model by calculating the empirical MMD. In other words, the procedure produces the same result even if one permute $n$ estimated conditional distributions $\{(\hat{\mu}(X_i), \hat{\sigma}(X_i))\}$ arbitrarily. This can lead to a failure in predicting the true conditional distribution even under the Gaussian condition, especially when they use multi-layer neural networks which are well-known universal approximators that can fit any continuous function [Hornik et al., 1989].

Zhao et al. [2020] try to give a randomized prediction of the cdf (in their paper denoted by $h[x, r](\cdot)$, where $r$ is some $\mathrm{Unif}[0, 1]$ random variable). The idea behind this is that if one gets a good cdf estimation, then $\hat{F}_x(Y)$ should be close to the uniform distribution. So the target is to minimize the empirical absolute difference between $h[x, r](y)$ and $r$ (denoted by $L_1$). If $L_1 < \epsilon$ with probability at least $1 - \delta$, then they call it $(\epsilon, \delta)$-probably approximately individual calibrated (PAIC). But minimizing $L_1$ alone cannot avoid being trapped by a naïve and meaningless calibration: $h[x, r](y) \equiv r$. So they combine the loss with another maximum likelihood type loss $L_2$ (negative logarithm of likelihood, by assuming the underlying distribution of $Y$ is Gaussian). But combining that loss cannot guarantee a meaningful calibration. To illustrate this, one just needs to consider the continuous distribution case, where each $X_i = x_i$ is only observed once almost surely. A global minimizer for the empirical loss $L_1 = \sum_{i=1}^n \frac{1}{n}|h[x_i, r_i](y_i) - r_i|$ is obtained if one predicts $h[x_i, r_i](\cdot)$ to be $\mathcal{N}(y_i - \sigma\Phi^{-1}(r_i), \sigma^2)$, where $\mathcal{N}$ is the Gaussian and $\Phi(\cdot)$ is the cdf of standard Gaussian. It can be easily checked that this Gaussian distribution's $r_i^{\text{th}}$ quantile is exactly $y_i$). The maximum likelihood type loss $L_2$ is now $\frac{1}{2}\log(\sigma^2) + \frac{1}{n}\sum_{i=1}^n \Phi^{-1}(r_i)^2 = \frac{1}{2}\log(\sigma^2) + \text{constant}$. Minimizing it means that the $\sigma^2$ term will asymptotically go to zero, leading to an overfit and meaningless result of $h[x_i, r_i](\cdot) \to \delta_{y_i}$. Again, this overfitting problem can be exacerbated by the fact that they train the model via a neural network, which is commonly believed to be highly overparametrized that can fit into any data nowadays. We must also point out that although Zhao et al. [2020] claims that they prove the existence of an $(\epsilon, \delta)$-PAIC predictor, the existence is only guaranteed by the existence of another monotonically PAIC (where monotonicity requires $h[x, r](y)$ is monotonically increasing with respect to $r$) predictor (see their Theorem 1). But they do not provide the existence of any non-trivial mPAIC predictor other than the trivial solution $h[x, r](y) \equiv r$. Therefore, we think their argument is at least incomplete, if not suspicious.

For those existing methods that do not assume Gaussian distributions, unfortunately, they are still inconsistent from an individual calibration perspective [Kuleshov et al., 2018, Chung et al., 2021].

Kuleshov et al. [2018] develop an isotonic regression way to calibrate the quantiles in the regression setting, which is inspired by the widely used isotonic regression in the classification setting, hence is a recalibration method based on some initial estimation of all quantiles (say $\{\tilde{Q}\}_\tau = \hat{F}^{-1}(\tau)$).

Then the authors propose an isotonic regression method that learns a monotonically increasing function $g(\tau_{\text{expected}}) = \tau_{\text{observed}}$, where $\tau_{\text{observed}} = \frac{1}{n}\sum_{i=1}^{n}\mathbb{1}\{Y_i \leq \tilde{Q}_{\tau_{\text{expected}}}\}$. The final output is $\hat{Q}_\tau = \tilde{Q}_{g^{-1}(\tau)}$. The major drawback of their method is that it heavily relies on the initial calibration $\tilde{Q}_\tau$. Hence the final marginal (or individual) recalibration result is only valid when the function class $\{\tilde{Q}_\tau : \tau \in [0,1]\}$ contains a marginal (or individual) calibrated prediction. Despite the marginal case, such a realizability condition is too strong to obtain in the individual calibration set, leading to the failure of their recalibration methods in the individual case.

Chung et al. [2021] propose a combined loss in their Section 3.2 with respect to both average calibration and sharpness. The first term minimizes the calibration error: $L_1 = \mathbb{1}\{\frac{1}{n}\sum_{i=1}^{n}\mathbb{1}\{Y_i \leq \hat{Q}_\tau\} < \tau\} \cdot \frac{1}{n}\sum_{i=1}^{n}[(Y_i - \hat{Q}_\tau(X_i))\mathbb{1}\{Y_i > \hat{Q}_\tau(X_i)\}] + \mathbb{1}\{\frac{1}{n}\sum_{i=1}^{n}\mathbb{1}\{Y_i \leq \hat{Q}_\tau\} > \tau\} \cdot \frac{1}{n}\sum_{i=1}^{n}[(-Y_i + \hat{Q}_\tau(X_i))\mathbb{1}\{Y_i < \hat{Q}_\tau(X_i)\}]$. Note that $L_1$ is zero if the prediction $\hat{Q}_\tau$ is marginally calibrated. The second term is $L_2 = \frac{1}{n}\sum_{i=1}^{n}\hat{Q}_\tau(X_i) - \hat{Q}_{1-\tau}(X_i)$, called sharpness regularization term. But that combined loss $L = (1 - \lambda)L_1 + \lambda L_2$ will fail even for calibrating such a simple instance: $X \sim \text{Unif}[0,1]$, $u \sim \text{Unif}[-1,1]$, and $Y = uX$. Any $\tau^{\text{th}}(\tau > 0.5)$ quantile should be $Q_\tau(F_x) = (\tau - 0.5)x$, while minimizing their proposed loss for $\lambda \in [0, \lambda_0]$ ends up with $\hat{Q}_\tau = x$ for those $x < \tau - 0.5$ and $\hat{Q}_\tau = 0$ for those $x \geq \tau - 0.5$. For larger $\lambda \in [\lambda_0, 1]$, minimizing their proposed loss will lead to an all-zero meaningless prediction $\hat{Q}_\tau = 0$. The detailed calculation can be found in Appendix.

Finally, we point out another issue that using the training residuals as errors to estimate the error distribution will lead to a biased (usually over-confident) estimation. For example, the Section 3.1 of Chung et al. [2021] proposes a combined method of first training a regression model $\hat{f}$ on $\{(X_i, Y_i)\}_{i=1}^{n}$ and then train a regression model on $\text{res}_i = Y_i - \hat{f}(X_i)$. But they use the same set of data to do both steps of training, leading to a statistically inconsistent estimator. The inconsistency becomes especially severe for modern highly overparametrized machine learning models that can fit into even random data [Zhang et al., 2021].

*Remark* 1. Many researchers [Chung et al., 2021, Zhao et al., 2020] argue that combining a sharpness regularization term with the average calibration loss will help to force the calibration model to stay away from marginal calibration. However, in this section, we show that those combining methods fail to provide a meaningful individually calibrated quantile prediction. In conclusion, a sharpness regularization term may help to achieve a marginal calibration, while it does no good for the goal of individual calibration.

## C  Discussions on Negative/Impossibility Results for Individual Calibration

Before elaborating on those impossibility results, we adopt the lower bound result from the nonparametric (mean) estimation literature Györfi et al. [2002], showing that any estimator may suffer from an arbitrarily slow convergence rate without additional assumptions on the underlying estimation, even in a noiseless setting. The original result is established for the conditional mean regression, and the same result holds for the conditional quantile estimator in our case.

**Theorem 6** (Theorem 3.1 in Györfi et al. [2002], rephrased). *For any sequence of conditional quantile estimators $\{\hat{Q}_{\tau,n}\}$ and any arbitrarily slow convergence rate $\{a_n\}$ with $\lim_{n\to\infty} a_n = 0$ and $a_1 \geq a_2 \geq \cdots > 0$, $\exists \mathcal{P}$, s.t. $X \sim \text{Unif}[0,1]$, $U|X \sim \delta_{g(X)}$ (hence $Q_\tau(U|X = x) = g(x), \forall \tau > 0$), where $g(X) \in \{-1, +1\}$, and*

$$\limsup_{n\to\infty} \frac{\mathbb{E}\left[\left|\hat{Q}_{\tau,n}(X) - g(X)\right|^2\right]}{a_n} \geq 1.$$

Such a negative result further justifies our Assumption 1 in a way that one cannot achieve any meaningful convergence rate guarantee without making any assumptions on the underlying distribution.

Discussions on Zhao et al. [2020]: The negative result obtained by Zhao et al. [2020] (their Proposition 1) is for the target of estimating the conditional distribution's cdf. The authors argue that the quality of an estimation $\hat{F}_x(\cdot)$ should be measured via how close the distribution $\hat{F}_x(Y)$ is to the uniform distribution $\text{Unif}[0,1]$. The measurement of closeness is defined by the Wasserstein-1 distance.

Their counter-example is constructed in a way that $\mathcal{P}_{Y|X=x}$ is a singleton (say, a delta distribution on the point $g(x)$, where $g(x)$ is chosen adversarially by the nature). They prove that any learner cannot distinguish between the true distribution and the adversarially chosen distribution from the observed data, thus any algorithm that claims a small closeness between $\hat{F}_x(Y)$ and $\mathrm{Unif}[0,1]$ cannot guarantee that the observed data are not sampled from the adversarial distribution. Since the Wasserstein-1 distance between a delta distribution and $\mathrm{Unif}[0,1]$ is at least $\frac{1}{4}$, the closeness claim is false for at least the adversarial distribution. We need to note that the measurement they take is a little suspicious: if the conditional distribution itself is a delta distribution, then the closeness of $F_x(Y)$ to the uniform distribution cannot be regarded as a good measurement of distribution calibration, since even the oracle will also suffer an at least $\frac{1}{4}$ Wasserstein distance. Although a certain negative result when $Y|X \sim \delta_{g(X)}$ is also made in our Theorem 6, we need to note that the closeness of $\hat{F}_x(Y)$ to $\mathrm{Unif}[0,1]$ can only be a good calibration measurement if the conditional distribution is continuous. On the contrary, for the general continuous conditional distributions (with only those mild assumptions), our methods can achieve a minimax optimal rate individual calibration guarantee with respect to any quantile (Theorem 1).

Discussions on Lei and Wasserman [2014], Vovk [2012]: Vovk [2012]'s impossibility result (their Proposition 4) is a version of Lemma 1 in Lei and Wasserman [2014]. Thus we will only discuss Lemma 1 in Lei and Wasserman [2014] here. The impossibility result is established for those distributions which have continuous marginal distributions $\mathcal{P}_X$. They focus on a conditional coverage condition, meaning that one should obtain a $(1-p)$ coverage set $\hat{C}(X_{\text{test}})$ such that $\mathbb{P}(Y_{\text{test}} \in \hat{C}(X_{\text{test}})|X_{\text{test}} = x) \geq 1 - p$ for any $x \in \mathcal{X}$ almost surely. Lei and Wasserman [2014] prove that for any distribution $\mathcal{P}$ and its any continuous point $x_0$, for any *finite* $n$, such an *almost sure* conditional coverage must result in an infinite expected length of covering set with respect to Lebesgue measure. This negative result seems to be disappointing for individual calibration at first glance. However, an almost sure guarantee in a finite sample case is usually too strong in practice. Our results are established for two weaker but more practical settings: either the strong consistency in the asymptotic case as $n \to \infty$ or a mean squared error guarantee at the finite sample case. A similar spirit of relaxing the requirement in order to get a positive result can also be found in Lei et al. [2018], where the authors get an asymptotic result that the covering band will converge to the oracle under certain assumptions. Note that their requirement is to get the *sharpest* covering band, meaning that they need to estimate the probability density function, which leads to their more restricted assumptions and different analysis, compared to ours.

## D  Comparison with Other Results

In this subsection, we make a comparison of the technical contributions with other related literature such as concentration of order statistics and Kernel Density Estimation (KDE).

Current quantile concentration analyses are all based on the i.i.d. assumption, which is not applicable to the continuous case conditional quantile estimation problem, where the learner almost surely does not observe identical features. But to complete our discussions, we give a brief introduction to the existing quantile concentration methods. There are two lines of research, which we call vertical and horizontal correspondingly. Vertical concentration guarantee is based on curving the convergence rate of the Smirnov process (the process of the empirical cdf converges to the true cdf) and applying the Dvoretzky-Kiefer-Wolfowitz inequality [Dvoretzky et al., 1956, Massart, 1990] to get a sharp coverage along the vertical line (in regards to cdf function value), that is, the get a sharp $\delta > 0$ such that $\hat{Q}_{\tau-\delta} \leq Q_\tau \leq \hat{Q}_{\tau+\delta}$ with high probability [Szorenyi et al., 2015, Altschuler et al., 2019]. But turning such a vertical guarantee that guarantees the coverage probability into a horizontal one that guarantees the estimation error is highly non-trivial. By assuming the cdf to be strictly increasing and continuous, Szorenyi et al. [2015], Torossian et al. [2019], Kolla et al. [2019], Howard and Ramdas [2022] get a horizontal concentration of order $\tilde{O}(1/\sqrt{n})$. Cassel et al. [2018] assume a Lipschitz cdf (hence an upper bound for pdf) and get a rate of $\tilde{O}(1/\sqrt{n})$. Altschuler et al. [2019] directly assumes a function (which they call $R$) of the cdf to transform the vertical concentration into a horizontal one. Another line of quantile concentration is to directly get a horizontal guarantee. Yu and Nikolova [2013] use Chebyshev's inequality to prove the horizontal concentration while remaining coarse-grained compared to Hoeffding type guarantee. Zhang and Ong [2021] utilize an alternative assumption and analysis of Boucheron and Thomas [2012] that relates to a certain

type of distribution with non-decreasing hazard rate (or monotone hazard rate, MHR). Although the MHR assumption is common in survival analysis, such an assumption is no longer appropriate for many other distributions, such as Gaussian. In our proof, we make a combination of the finite sample theory of the M-estimator and the local strong convexity of the conditional expected pinball loss to derive a new proof for the concentration of empirical conditional quantile around the true conditional quantile. We do not assume the upper bound or lower bound on the pdf, the strict increasing cdf, or the non-decreasing hazard rate. We also generalize the i.i.d. setting of quantile concentration to the independent but not identical setting, which is the first one to our knowledge in the quantile concentration literature.

Another important line of research is the kernel density estimation (KDE) literature focuses on giving a full characterization of the conditional distribution. Such a goal is aligned with the need for the full characterization of pdf in some downstream tasks. For example, if one wants to find the sharpest covering set (or covering band, if one further assumes the conditional distribution is univariate), then one must identify the probability density function so that thresholding can thus be constructed [Lei and Wasserman, 2014, Lei et al., 2018]. But for a simpler and probably wider class of downstream tasks, such as decision-making based on quantiles (newsvendor problem) and evaluating and minimizing the quantiles (robust optimization), an accurate estimation of the conditional quantiles should suffice.

From the point of developing practical algorithms, KDE methods are often computationally intractable for conditional quantile estimations. To get a quantile estimation, one has to integrate the full probability density function. The whole procedure is highly costly even if one discretized the integration. Another point is that theoretical guarantees in the KDE literature are often established with respect to the mean integrated squared error (MISE), while it does not simultaneously guarantee convergence with respect to some quantile. Consider a simple instance where the estimated probability density function $\hat{p}(y)$ differs from true probability density function $p(y)$ by a factor of $O\left(\frac{1}{\sqrt{n}} \cdot \frac{1}{|y|}\right)$. The MISE converges with a rate of $O\left(\frac{1}{n}\right)$. However, the cumulative pdf estimation error (denoted by $\int_{-\infty}^{y} |\hat{p}(t) - p(t)| \mathrm{d}t$ may diverge (since $\int_{-\infty}^{0} \frac{1}{|t|} \mathrm{d}t$ is a divergent integration). The divergence is caused by the non-exchangeability between the square and the integration.

Since the final goal is to get an estimation of the conditional quantile, we can relax the assumptions in the KDE literature largely. For those nonparametric methods such as Lei and Wasserman [2014], they require the smoothness (to be more specific, Hölderness) of the underlying conditional pdf with respect to $y$ and the Lipschitzness with respect to $x$ in the infinite norm, which is a far stronger assumption than our case. Instead, we only assume that the conditional quantile (rather than the full pdf with respect to the sup norm) is Lipschitz. For those methods that establish the KDE convergence results in a generalization bound via complexity arguments, our assumption is still much weaker. For example, Bilodeau et al. [2021] make a realizability assumption (that the true pdf must lie in the hypothesis class) to achieve a generalization bound, while our assumption just requires the true conditional quantile lies in a bounded set $[-M, M] \subset \mathbb{R}$.

In a word, our analysis combines the idea of decomposing the error into bias and variance terms as in nonparametric regression methods and the spirit of the covering number arguments in the analysis of parametric models. Such a combination is novel and non-trivial, which is critical for the relaxation of the assumptions.

# E  Numerical Experiment Details

In this part, we provide detailed experiment settings for Section 4.

## E.1  Evaluation metrics

- Mean Absolute Calibration Error (MACE): for (expected) quantile level $\tau$ and a quantile prediction model $Q(X, \tau)$, define "observed quantile level" by

$$\tau^{obs} := \frac{1}{n} \sum_{i=1}^{n} \mathbb{1}\{Y_i \leq Q(X_i, \tau)\}.$$

For pre-specified quantile levels $0 \leq \tau_1 < \cdots < \tau_M \leq 1$, MACE is calculated by:

$$\text{MACE}(\{(X_i, Y_i)\}_{i=1}^n, Q) := \frac{1}{M} \sum_{j=1}^M \left| \tau_j - \tau_j^{obs} \right|.$$

In practice, we set $M = 100$ and pick $\tau$'s from 0.01 to 0.99 with equal step size.

- Adversarial Group Calibration Error (AGCE): introduced in Zhao et al. [2020], AGCE is calculated in practice by first sub-sampling (possibly with replacement) $B_1, \cdots, B_r \subseteq \{(X_i, Y_i)\}_{i=1}^n$ and then retrieve the maximum MACE from the pool:

$$\text{AGCE}(\{B_k\}_{k=1}^r, Q) := \max_k \{\text{MACE}(B_k, Q)\}.$$

- Check Score: check score is an empirical version of the expected pinball loss. For pre-specified quantile levels $0 \leq \tau_1 < \cdots < \tau_m \leq 1$ (in our experiments $\tau$'s are set in the same way as in MACE), the check score is defined as:

$$\text{CheckScore}(\{(X_i, Y_i)\}_{i=1}^n, Q) := \frac{1}{m} \sum_{j=1}^m \rho_{\tau_j}(Q(X, \tau_j), Y),$$

with $\rho_\tau(Q(X, \tau), Y) := \frac{1}{n} \sum_{i=1}^n l_\tau(Q(X_i, \tau), Y_i)$.

- Length: our length metric gives the average interval length of a predicted 90% confidence interval. The interval is given as the area between the lower quantile (at level $\tau_{\text{low}} = 0.05$) and the upper quantile (at level $\tau_{\text{up}} = 0.95$):

$$\text{Length}(\{(X_i, Y_i)\}_{i=1}^n, Q) := \frac{1}{n} \sum_{i=1}^n (Q(X, \tau_{\text{up}}) - Q(X, \tau_{\text{low}})).$$

- Coverage: to encourage a fairer comparison, we always attach the empirical coverage rate to the length metric. Coverage rate evaluates the percentage of real data that is covered by the given 90% confidence intervals:

$$\text{Coverage}(\{(X_i, Y_i)\}_{i=1}^n, Q) := \frac{1}{n} \sum_{i=1}^n \mathbb{1}\{Q(X_i, \tau_{\text{low}}) \leq Y_i \leq Q(X_i, \tau_{\text{up}})\}.$$

We make a brief explanation for the selection of the performance metrics as follows. The former three metrics are typical measurements in the literature of quantile regression and uncertainty calibration that measure the quality of the calibrations, while the latter two are classic metrics in the field of conformal prediction. Although the quantile calibration task is very closely related to the conformal prediction task, the goals are still different. Conformal prediction aims to provide as sharp as possible prediction sets that cover the true outcomes with desired rates. While a precise quantile prediction guarantees the coverage rate, such a covering band may not be the sharpest. As is discussed in Appendix B, the additional sharpness regularization term could harm the goal of precise quantile prediction. For example, people may select 0.05 and 0.95 quantiles to construct a 0.9 coverage band, but the sharpest covering band probably would deviate from such quantiles. However, a high-quality quantile prediction alone is of independent interest for many downstream tasks, such as the newsvendor problem. This points to a difference between the objectives of calibration and conformal prediction. As a result, the popular performance metrics applied in conformal prediction such as the average interval length and the coverage rate are not direct measurements for the quantile calibration problem. Nevertheless, we still present the Length and the Coverage metrics for completeness. Some other measurements, for example, the independence between the coverage indicator and the band length in Feldman et al. [2021], are only a necessary condition of the quantile calibration and are therefore not considered here.

## E.2 Synthetic Dataset

For an example of heteroscedastic Gaussian noises, consider an underlying generation scheme of $(X, Y) \in \mathbb{R}^2$ from $X \sim \text{Unif}[0, 15]$ and $Y|X \sim \mathcal{N}(\mu_0(X), \sigma_0^2(X))$, with $\mu_0(X) = 4 \cdot \sin(\frac{2\pi}{15} X)$ and $\sigma_0(X) = \max\{0.2X \cdot |\sin(X)|, 0.1\}$. We generate $n_{\text{train}} = 40000$ samples for training and

$n_{\text{test}} = 1000$ for testing. For our NRC method, we randomly select $n_1 = 36000$ for training a regression model and $n_{\text{train}} - n_1 = 4000$ samples for the quantile calibration step. All neural networks used in this example have hidden layers of size $[100, 50]$. As shown in Figure 1, our NRC method successfully captures the actual fluctuation of conditional variance, while the traditional benchmark model Deep Ensemble which relies on 5 heteroscedastic neural networks cannot capture the individual dependence on the feature of the noise variance.

### E.3    Dimension reduction implementation

As given in Theorem 1, for high dimensional data (i.e., large $d$) the upper bound guarantee may get too loose as a theoretical guarantee. Motivated by this dilemma, we propose applying dimension-reduction techniques when doing nonparametric approximation. We summarize them as the NRC-DR algorithms. There are loads of choices for dimension reduction, in this work we only implement three of them: random projection, covariate selection, and second-to-last layer feature extraction.

- Random Projection Dasgupta [2013]: we use the Gaussian random projection method, in which we project the $d$-dimensional inputs on the column space of a randomly generated $d_0 \times d$ matrix (assuming that $d_0 < d$, otherwise original input will be retained), whose elements are independently generated from $\mathcal{N}(0, \frac{1}{d})$.

- Covariate Selection: we select the most relevant features from the input vector. In this experiment, we select $d_0$ out of $d$ features that have the largest absolute value of Pearson correlation coefficient with the target variable.

- Second-to-Last Layer Embedding: The second-to-last layer of a trained neural network has been widely used for feature extraction when the input dimension is high. We add an additional layer of size $d_0$ to the regression network $\hat{f}(\cdot)$ in Algorithm 2 and directly use this layer as the feature embedding.

NRC algorithms implemented with each of these dimension-reduction techniques are called NRC-RP, NRC-Cov, and NRC-Embed respectively. In the UCI-8 experiments we implement NRC-RP and NRC-Cov, with the full result presented in Table 4, while for high-dimensional datasets we choose the medical expenditure panel survey (MEPS) datasets [panel 19, 2017, panel 20, 2017, panel 21, 2017] as suggested in [Romano et al., 2019] to test out our NRC-RP and NRC-Embed variants. The sizes and feature dimensions of these datasets are listed in Table 3. Due to the relative abundance of training data, we set the hidden layers of neural networks to $[64, 64]$, the rest hyperparameters are pre-determined by grid search. For the results presented in Table 2, we reduce the dimension to $d_0 = 30$.

### E.4    Benchmark methods

We explain the benchmark models and uncertainty quantification methods we are competing against in Section 4. The first type of method is based on the Gaussian assumption. They assume that the conditional distribution of $Y$ given $X$ can be fully determined by the conditional mean and variance. These models are of the same output structure of two neurons: one for prediction of $\mu(X)$, another for $\sigma(X)$. At the implementation of the experiments, we assume that $\mu(X)$ and $\sigma(X)$ share the same overall network structure, with two hidden layers of size $[20, 20]$ for small datasets and $[64, 64]$ for large or high dimensional datasets. Amongst them, MC-Dropout (MCDrop) Gal and Ghahramani [2016], Deep Ensembles (DeepEnsemble) Lakshminarayanan et al. [2017] and Heteroscedastic Neural Network (HNN) Kendall and Gal [2017] are trained by minimizing the negative log-likelihood (NLL) error. The Maximum Mean Discrepancy method (MMD) Cui et al. [2020] trains by first minimizing the NLL loss of an HNN model, and then re-train the model to minimize the MMD between the model prediction and the same training set at the second phase. The method introduced in section 3.2 of Chung et al. [2021] introduces a combined calibration loss (CCL) with an average calibration loss and a sharpness regularization term, and the model is trained by minimizing it. Apart from these feed-forward structures, there is also a Deep Gaussian Process model (DGP) Damianou and Lawrence [2013] which assumes that the data are generated from a Gaussian process, and is trained by the doubly stochastic variational inference algorithm Salimbeni and Deisenroth [2017]. The DGP model is of high training cost, we thus only test it on small datasets. We also run experiments on post-hoc calibration methods, such as isotonic regression (ISR) suggested by Kuleshov et al. [2018]. In our experiments, we evaluate the post-hoc ISR method by attaching it

to a pre-trained HNN model. We also implement another post-hoc method named Model Agnostic Quantile Regression (MAQR) method given in section 3.1 of Chung et al. [2021]. Their algorithm is similar to ours in terms of training a regression model first, but different in two aspects: first, MAQR uses the same dataset to do both regression training and quantile calibration; second, MAQR obtains a quantile result at the second phase via training a new quantile regression model, which in our implementation is a neural network. For the algorithms coming from the conformal prediction society, we test the Conformalized Quantile Regression (CQR) method introduced by Romano et al. [2019] and the Orthogonal Quantile Regression (OQR) approach from Feldman et al. [2021]. The CQR algorithm also takes a training step followed by a calibration step, with the training step learning the target quantiles by minimizing the pinball loss and the calibration step further calibrating the quantile predictions by conformal prediction. Based on the CQR paradigm, the OQR algorithm further improves *conditional coverage* by introducing an additional regularization term to the vanilla pinball loss used for quantile regression. The regularization term encourages the independence between interval length and the violation (falling out of the predicted interval) event of the response variable.

### E.5   Experiment details on the 8 UCI datasets

We have run extensive experiments to compare our methods against multiple state-of-the-art benchmark models. The full results are listed in Table 4. A brief review of all the benchmarks has been provided in Section E.4, and realizations of algorithms NRC-RP and NRC-Cov are explained in Appendix E.3.

A short summary of the 8 UCI datasets is listed in 3. For all the network structures used in Section 4 we fix the size of the hidden layers to $[20, 20]$ (and $[10]$ for the DGP model). For our dimension reduction algorithms, the target dimension that we reduce to is set to 4. The rest hyperparameters (including the learning rate, minibatch size, kernel width, etc) are pre-determined by running a grid search. The learning rate is searched within $[10^{-2}, 5 \times 10^{-3}, 10^{-3}]$, the minibatch size is searched within $[10, 64, 128]$, and the rest hyperparameters exclusive to each model are searched in the same fashion. For each possible combination of model and dataset, we save its optimal hyperparameters into separate configuration files.

During the main experiment, for each combination of model and dataset, we load its optimal hyperparameter configurations, on which we repeat the whole training process 5 times. Each time we randomly split out $10\%$ of the whole sample as the testing set. On the remaining set, for our NRC algorithms we additionally separate out $30\%$ for recalibration, and for the rest algorithms, no additional partitioning is required. The rest data is then fed into the whole train-validation loop, and we set up an early stopping scheme with a patience count of 20. That is, if for 20 epochs no decrease in loss is observed on the validation set, then the training is early stopped.

Table 3: Descriptions of Datasets. Each row gives the number of examples contained in the corresponding dataset and the number of features in the input variable.

(a)

| Dataset | # Examples | # Features |
|---|---|---|
| Boston | 506 | 13 |
| Concrete | 1030 | 8 |
| Energy | 768 | 8 |
| Kin8nm | 8192 | 8 |
| Naval | 11934 | 17 |
| Power | 9568 | 4 |
| Wine | 4898 | 11 |
| Yacht | 308 | 6 |

(b)

| Dataset | # Examples | # Features |
|---|---|---|
| Meps_19 | 15785 | 139 |
| Meps_20 | 17541 | 139 |
| Meps_21 | 15656 | 139 |
| Bike-Sharing | 17379 | 11 |
| Wine-Red | 1599 | 11 |
| Auto-MPG | 392 | 6 |

Table 4: Full table of experiment results on 8 UCI datasets

| Dataset | Metric | HNN | MCDrop | DeepEnsemble | CCL | ISR | DGP | MMD |
|---|---|---|---|---|---|---|---|---|
| Boston | MACE | 0.065±0.04 | 0.14±0.03 | 0.06±0.04 | 0.11±0.04 | 0.057±0.03 | 0.11±0.01 | 0.18±0.03 |
| | AGCE | 0.25±0.09 | 0.22±0.04 | 0.27±0.07 | 0.34±0.1 | 0.27±0.08 | 0.36±0.1 | 0.41±0.09 |
| | CheckScore | 0.96±0.2 | 1.9±0.2 | 1±0.3 | 1.1±0.4 | 0.95±0.2 | 1.1±0.3 | 1.3±0.4 |
| | Length(Coverage) | 8.9±1(88%) | 8.6±1(95%) | 8.5±1(90%) | 5.3±0.7(90%) | 7.9±1(88%) | 4.6±0.1(75%) | 10±1(97%) |
| Concrete | MACE | 0.038±0.02 | 0.078±0.02 | 0.053±0.03 | 0.11±0.03 | 0.044±0.02 | 0.17±0.02 | 0.098±0.05 |
| | AGCE | 0.18±0.03 | 0.2±0.04 | 0.18±0.03 | 0.22±0.03 | 0.19±0.05 | 0.25±0.04 | 0.24±0.07 |
| | CheckScore | 2.2±0.6 | 3.6±0.2 | 1.9±0.3 | 2.3±0.6 | 2.3±0.6 | 1.7±0.2 | 3.9±1 |
| | Length(Coverage) | 19±1(89%) | 25±4(95%) | 19±1(93%) | 17±1(77%) | 20±0.8(89%) | 5.1±0.1(50%) | 23±3(94%) |
| Energy | MACE | 0.07±0.04 | 0.16±0.02 | 0.065±0.04 | 0.13±0.07 | 0.055±0.03 | 0.085±0.01 | 0.13±0.05 |
| | AGCE | 0.2±0.07 | 0.21±0.01 | 0.19±0.04 | 0.29±0.1 | 0.21±0.07 | 0.2±0.03 | 0.28±0.1 |
| | CheckScore | 0.61±0.1 | 1.4±0.3 | 0.64±0.09 | 0.79±0.1 | 0.6±0.1 | 0.25±0.02 | 0.71±0.2 |
| | Length(Coverage) | 7.8±2(88%) | 9±4(93%) | 7.7±0.6(94%) | 2.1±0.2(60%) | 7.1±2(95%) | 4.2±0.4(97%) | 8±0.8(92%) |
| Kin8nm | MACE | 0.049±0.04 | 0.034±0.01 | 0.077±0.05 | 0.1±0.02 | 0.013±0.005 | 0.035±0.01 | 0.13±0.01 |
| | AGCE | 0.11±0.04 | 0.075±0.01 | 0.12±0.06 | 0.13±0.02 | 0.072±0.02 | 0.08±0.03 | 0.18±0.02 |
| | CheckScore | 0.031±0.001 | 0.045±0.002 | 0.027±0.002 | 0.03±0.002 | 0.031±0.001 | **0.023±0.001** | 0.035±0.003 |
| | Length(Coverage) | 0.26±0.005(88%) | 0.5±0.02(95%) | 0.29±0.006(93%) | 0.25±0.003(80%) | 0.28±0.007(90%) | 0.28±0.002(94%) | 0.37±0.002(86%) |
| Naval | MACE | 0.12±0.05 | 0.1±0.03 | 0.12±0.05 | 0.18±0.1 | **0.011±0.005** | 0.12±0.06 | 0.2±0.007 |
| | AGCE | 0.17±0.06 | 0.15±0.05 | 0.14±0.06 | 0.22±0.1 | 0.054±0.009 | 0.15±0.05 | 0.2±0.005 |
| | CheckScore | 0.00086±0.0002 | 0.0018±0.0006 | **0.00064±0.0002** | 0.0021±0.0006 | 0.00079±0.0003 | 0.0026±0.0006 | 0.006±0.0007 |
| | Length(Coverage) | 0.013±0.0008(99%) | 0.02±0.002(94%) | 0.015±0.0003(100%) | 0.016±0.0007(84%) | 0.0078±0.0008(90%) | 0.05±0.004(97%) | 0.0054±0.0003(77%) |
| Power | MACE | 0.045±0.02 | 0.29±0.01 | 0.045±0.03 | 0.16±0.07 | 0.011±0.006 | 0.13±0.01 | 0.21±0.03 |
| | AGCE | 0.1±0.02 | 0.3±0.01 | 0.075±0.03 | 0.21±0.08 | 0.065±0.02 | 0.16±0.007 | 0.23±0.03 |
| | CheckScore | 1.3±0.09 | 24±3 | 1.3±0.1 | 1.5±0.2 | 1.3±0.1 | 1.3±0.009 | 1.7±0.2 |
| | Length(Coverage) | 13±1(87%) | 20±3(97%) | 16±0.5(97%) | 14±1(80%) | 13±0.8(90%) | 17±0.1(86%) | 13±1(85%) |
| Wine | MACE | 0.05±0.04 | 0.036±0.02 | 0.049±0.03 | 0.057±0.009 | 0.016±0.009 | 0.047±0.03 | 0.13±0.02 |
| | AGCE | 0.11±0.05 | 0.1±0.03 | 0.12±0.05 | 0.12±0.03 | 0.091±0.02 | 0.11±0.05 | 0.16±0.02 |
| | CheckScore | 0.2±0.01 | 0.2±0.01 | 0.2±0.009 | 0.21±0.01 | 0.2±0.01 | 0.2±0.01 | 0.24±0.02 |
| | Length(Coverage) | 2.4±0.07(89%) | 2.4±0.09(90%) | 2.2±0.02(88%) | 2.3±0.06(86%) | 2.2±0.03(87%) | 2.3±0.008(89%) | 2.4±0.08(92%) |
| Yacht | MACE | 0.075±0.05 | 0.11±0.03 | **0.049±0.02** | 0.11±0.04 | 0.06±0.03 | 0.11±0.03 | 0.14±0.05 |
| | AGCE | 0.34±0.1 | 0.29±0.08 | 0.35±0.06 | 0.45±0.05 | 0.34±0.09 | **0.29±0.05** | 0.42±0.08 |
| | CheckScore | 1.4±0.6 | 1.9±0.6 | 1.1±0.4 | 0.49±0.2 | 1.5±0.6 | 0.54±0.2 | 1.1±0.3 |
| | Length(Coverage) | 8.1±2(70%) | 18±5(94%) | 10±1(91%) | 7.8±1(86%) | 10±0.9(88%) | 5.3±0.2(85%) | 9.2±2(90%) |

| Dataset | Metric | MAQR | CQR | OQR | NRC | NRC-RF | NRC-RP | NRC-Cov |
|---|---|---|---|---|---|---|---|---|
| Boston | MACE | 0.051±0.02 | 0.049±0.02 | 0.057±0.03 | 0.055±0.02 | **0.048±0.02** | 0.055±0.02 | 0.055±0.02 |
| | AGCE | 0.25±0.06 | 0.24±0.06 | 0.19±0.05 | 0.19±0.05 | 0.25±0.08 | 0.26±0.07 | **0.19±0.05** |
| | CheckScore | 1.1±0.2 | **0.72±0.1** | 0.73±0.2 | 1.1±0.2 | 1.2±0.4 | 1.1±0.2 | 1.1±0.2 |
| | Length(Coverage) | 9.1±1(88%) | 7.1±0.7(82%) | 8±1(90%) | 10±1(94%) | 9.7±1(91%) | 11±1(94%) | 9.6±1(90%) |
| Concrete | MACE | 0.049±0.03 | 0.059±0.03 | 0.048±0.02 | 0.058±0.02 | 0.047±0.02 | **0.038±0.03** | 0.045±0.02 |
| | AGCE | 0.18±0.06 | 0.23±0.05 | 0.18±0.05 | 0.19±0.05 | **0.18±0.05** | 0.19±0.04 | 0.2±0.06 |
| | CheckScore | 2±0.2 | 1.72±0.6 | 1.57±0.2 | 1.9±0.4 | **1.4±0.2** | 2.6±0.7 | 2±0.4 |
| | Length(Coverage) | 17±2(79%) | 20±0.7(89%) | 17±0.8(89%) | 21±0.8(90%) | 19±0.9(92%) | 21±1(89%) | 21±1(89%) |
| Energy | MACE | 0.063±0.02 | 0.075±0.03 | 0.058±0.01 | 0.04±0.01 | **0.037±0.008** | 0.04±0.01 | 0.04±0.01 |
| | AGCE | 0.21±0.04 | 0.18±0.008 | **0.17±0.006** | 0.26±0.1 | 0.22±0.06 | 0.24±0.08 | 0.26±0.1 |
| | CheckScore | 0.6±0.2 | **0.14±0.03** | 0.31±0.2 | 0.69±0.2 | 0.18±0.03 | 0.69±0.2 | 0.69±0.2 |
| | Length(Coverage) | 6.2±3(84%) | 1.8±0.3(95%) | 3.7±0.8(94%) | 4.5±1(88%) | 1.5±0.1(89%) | 8.5±3(91%) | 7.3±2(91%) |
| Kin8nm | MACE | 0.032±0.02 | 0.041±0.02 | 0.053±0.01 | 0.017±0.007 | **0.013±0.004** | 0.017±0.007 | 0.017±0.007 |
| | AGCE | 0.072±0.02 | 0.075±0.02 | 0.099±0.01 | 0.07±0.02 | 0.067±0.02 | **0.056±0.02** | 0.07±0.02 |
| | CheckScore | 0.029±0.001 | 0.025±0.0009 | 0.036±0.001 | 0.032±0.002 | 0.044±0.001 | 0.032±0.002 | 0.032±0.002 |
| | Length(Coverage) | 0.27±0.01(87%) | 0.26±0.005(89%) | 0.45±0.005(93%) | 0.25±0.003(90%) | 0.52±0.007(90%) | 0.28±0.006(89%) | 0.27±0.009(90%) |
| Naval | MACE | 0.043±0.02 | 0.077±0.07 | 0.24±0.2 | 0.017±0.007 | 0.012±0.005 | 0.017±0.007 | 0.017±0.007 |
| | AGCE | 0.081±0.01 | 0.098±0.05 | 0.27±0.1 | **0.052±0.008** | 0.054±0.01 | 0.063±0.02 | 0.052±0.008 |
| | CheckScore | 0.0021±0.0007 | 0.0011±0.0009 | 0.0032±0.001 | 0.002±0.0007 | 0.00088±0.0001 | 0.002±0.0007 | 0.002±0.0007 |
| | Length(Coverage) | 0.017±0.003(84%) | 0.015±0.002(98%) | 0.023±0.004(63%) | 0.0097±0.002(91%) | 0.0088±0.0006(91%) | 0.012±0.002(90%) | 0.01±0.002(91%) |
| Power | MACE | 0.029±0.02 | 0.049±0.005 | 0.057±0.007 | **0.0086±0.003** | 0.014±0.008 | 0.01±0.004 | 0.0086±0.003 |
| | AGCE | 0.06±0.01 | 0.071±0.02 | 0.084±0.008 | 0.057±0.007 | 0.058±0.01 | **0.053±0.008** | 0.057±0.007 |
| | CheckScore | 1.2±0.03 | 1.1±0.03 | 1.2±0.08 | 1.2±0.05 | **1±0.03** | 1.3±0.07 | 1.2±0.05 |
| | Length(Coverage) | 12±0.8(88%) | 13±0.7(93%) | 14±1(95%) | 13±0.3(91%) | 11±0.1(90%) | 14±1(91%) | 13±1(92%) |
| Wine | MACE | 0.029±0.02 | 0.056±0.007 | 0.05±0.004 | 0.017±0.006 | **0.016±0.01** | 0.017±0.006 | 0.017±0.006 |
| | AGCE | 0.094±0.03 | 0.1±0.06 | 0.13±0.08 | **0.075±0.03** | 0.077±0.02 | 0.078±0.03 | 0.075±0.02 |
| | CheckScore | 0.2±0.02 | 0.21±0.02 | 0.21±0.02 | 0.21±0.01 | **0.19±0.01** | 0.21±0.01 | 0.21±0.01 |
| | Length(Coverage) | 2.3±0.1(84%) | 2.3±0.04(89%) | 2.3±0.03(89%) | 2.3±0.04(90%) | 2.2±0.02(90%) | 2.3±0.05(90%) | 2.4±0.06(90%) |
| Yacht | MACE | 0.088±0.04 | 0.073±0.05 | 0.14±0.08 | 0.12±0.06 | 0.096±0.05 | 0.087±0.04 | 0.093±0.05 |
| | AGCE | 0.31±0.07 | 0.34±0.04 | 0.38±0.05 | 0.34±0.07 | 0.42±0.08 | 0.31±0.08 | 0.31±0.04 |
| | CheckScore | 0.32±0.09 | 0.19±0.07 | 0.41±0.2 | 0.44±0.2 | **0.18±0.03** | 0.54±0.2 | 0.4±0.2 |
| | Length(Coverage) | 3.3±1(83%) | 1.7±0.4(72%) | 3.6±0.8(88%) | 5.3±1(93%) | 3.2±1(91%) | 4.5±1(93%) | 5.5±1(91%) |

## E.6 Experiment details on the Bike-Sharing dataset

The Bike-Sharing dataset is a time series dataset that records the number of bike rentals as well as the value of 11 related features at each hour. As briefly described in Section 4, for this experiment we design the LSTM-HNN and LSTM-NRC models, each having 2 layers, 40 hidden neurons for each layer, and a sliding window of size 5. For the LSTM-NRC model, in the recalibration step we are required to calculate (as part of the whole kernel matrix calculation) the distance between to input variables, and we retrieve the input vector by concatenating all the features within the history window (into a 55-dimension vector). As the dimension is high, we adopt dimension-reduction techniques and use random projection to map the concatenated feature vector to a 4-dimensional space.

Most experiment details, in terms of hyperparameter selection, early stopping criteria, etc, as similar to the settings described in Appendix E.5. One difference lies in the splitting of the dataset. Due to the temporal property, we are only allowed to predict "the future" from "the past", thus we split the starting 80% of time stamps for training (and further split for recalibration, if required), and the ending 20% of time stamps for testing.

### E.7 Robustness to Covariate Shift

Covariate shift [Quinonero-Candela et al., 2008] describes a frequently observed phenomenon that the distribution of the covariates (also called the independent variable, feature variable, etc) that differ between the training and testing datasets. In the presence of covariate shift, a calibration model attained from the training set that only achieves average calibration (i.e., naive predictions of quantiles) may no longer suit well for the testing dataset, while an individually calibrated model maintains its optimality in terms of all the metrics mentioned above.

Following the experiment design in Chen et al. [2016] and Wen et al. [2014], we experiment on both real datasets and half-synthetic ones (all of which are available on Dua and Graff [2017]). The experiment design is almost the same as in Section 4, except for a difference in the processing of testing sets. The experiment results on a subset of benchmarks are presented in Table 5, which further validates the robustness of our algorithm to covariate shift, especially from the dominating performance on the half-synthetic datasets. A few remarks on the datasets: the Wine dataset and Auto-MPG dataset both induce a natural interpretation of covariate shift, with the training set and the testing set having different colors in the experiment on the Wine dataset, and different origin cities on the Auto-MPG dataset; and on the rest two datasets we synthetically simulate a covariate shift for the testing set, as recommended in Chen et al. [2016]. To be specific, we introduce a covariate shift into the testing set by following steps:

1. Randomly split out $10\%$ of the whole dataset as the "testing pool".
2. On the training set $X_T$, Calculate empirical mean and variance $\hat{\mu} = \overline{X_T}, \hat{\Sigma} = \mathrm{Cov}(X_T)$.
3. Sample $X_{\mathrm{seed}} \sim \mathcal{N}(\hat{\mu}, \hat{\Sigma})$.
4. Resample (with replacement) 1000 examples from the testing pool, following density $\mathcal{N}(X_{\mathrm{seed}}, 0.3\hat{\Sigma})$.

Table 5: Experiments on covariate shift datasets. The first three rows are real datasets, while the last two are half-synthetic ones. For the Auto-MPG dataset, models are always trained on data samples from city 1, and tested on dataset of either city 2 or city 3, denoted by "MPG2" and "MPG3" respectively. Our algorithm has an outstanding performance, especially so on the two half-synthetic datasets.

| Dataset | Metric | HNN | MCDrop | DeepEnsemble | ISR | DGP | NRC |
|---------|--------|-----|--------|--------------|-----|-----|-----|
| Wine | MACE | 0.32±0.1 | 0.22±0.07 | 0.32±0.06 | 0.31±0.09 | 0.26±0.06 | **0.2±0.03** |
| | AGCE | 0.33±0.1 | 0.23±0.07 | 0.34±0.06 | 0.33±0.08 | 0.28±0.05 | **0.23±0.03** |
| | CheckScore | 0.51±0.2 | **0.29±0.06** | 0.43±0.1 | 0.48±0.2 | 0.31±0.06 | 0.31±0.09 |
| MPG12 | MACE | 0.065±0.03 | 0.12±0.01 | 0.07±0.03 | **0.059±0.02** | 0.17±0.01 | 0.069±0.008 |
| | AGCE | **0.21±0.05** | 0.22±0.04 | 0.19±0.03 | 0.21±0.06 | 0.30±0.05 | 0.29±0.06 |
| | CheckScore | 1.2±0.2 | 1.5±0.2 | **1.1±0.04** | 1.1±0.06 | 1.7±0.2 | 1.2±0.05 |
| MPG13 | MACE | 0.1±0.02 | 0.16±0.02 | 0.11±0.02 | 0.1±0.03 | 0.23±0.03 | **0.08±0.03** |
| | AGCE | 0.2±0.009 | 0.2±0.03 | **0.2±0.02** | 0.24±0.06 | 0.37±0.07 | 0.26±0.1 |
| | CheckScore | 1.1±0.04 | 1.6±0.3 | 1.2±0.08 | 1±0.05 | 1.4±0.07 | **0.9±0.06** |
| Concrete | MACE | 0.1±0.04 | 0.17±0.09 | 0.12±0.03 | 0.059±0.01 | 0.19±0.02 | **0.05±0.008** |
| | AGCE | 0.13±0.04 | 0.18±0.08 | 0.15±0.04 | 0.093±0.02 | 0.21±0.01 | **0.07±0.01** |
| | CheckScore | 4.11±0.2 | 4.8±0.8 | 4.3±0.2 | 4±0.3 | **2.3±0.27** | 2.5±0.1 |
| Boston | MACE | 0.24±0.08 | 0.33±0.1 | 0.2±0.01 | 0.11±0.02 | 0.08±0.03 | **0.05±0.02** |
| | AGCE | 0.26±0.07 | 0.34±0.1 | 0.21±0.01 | 0.14±0.02 | 0.1±0.03 | **0.075±0.02** |
| | CheckScore | 2.1±0.7 | 3.2±1.2 | 1.5±0.05 | 1.1±0.1 | 0.71±0.3 | **0.61±0.05** |

## F    Some Basic Probability Results

In this section, we present several well-known probability theory results.

**Lemma 1** (Bernstein's Inequality). *Let $\xi_1, \dots, \xi_n$ be $N$ independent random variables. Suppose $|\xi_i| \leq M_0$ almost surely. Then $\forall t > 0$,*

$$\mathbb{P}\left(\left|\frac{1}{N}\sum_{i=1}^{N}(\xi_i - \mathbb{E}[\xi_i])\right| \geq t\right) \leq \exp\left(-\frac{\frac{1}{2}Nt^2}{\sum_{i=1}^{N}\frac{1}{N}\mathrm{Var}(\xi_i) + \frac{1}{3}M_0 t}\right).$$

**Lemma 2** (Hoeffding's Inequality). *Let $\xi_1, \dots, \xi_n$ be $N$ independent random variables. Suppose $a_i \leq \xi_i \leq b_i$ almost surely. Then $\forall t > 0$,*

$$\mathbb{P}\left(\sum_{i=1}^{N}(\xi_i - \mathbb{E}[\xi_i]) \geq t\right) \leq \exp\left(-\frac{2t^2}{\sum_{i=1}^{N}(b_i - a_i)^2}\right),$$

$$\mathbb{P}\left(\sum_{i=1}^{N}(\mathbb{E}[\xi_i] - \xi_i) \geq t\right) \leq \exp\left(-\frac{2t^2}{\sum_{i=1}^{N}(b_i - a_i)^2}\right).$$

**Lemma 3** (Poisson Approximation). *Let $S_n = \sum_{k=1}^{n}\xi_{n,k}$, where for each $n$ the random variables $\xi_{n,k}, k \in [n]$ are mutually independent, each taking value in the set of non-negative integers. Suppose that $p_{n,k} = \mathbb{P}(\xi_{n,k} = 1)$ and $\epsilon_{n,k} = \mathbb{P}(\xi_{n,k} \geq 2)$ are such that as $n \to \infty$,*

*(a) $\sum_{k=1}^{n} p_{n,k} \to \lambda < \infty$;*

*(b) $\max_{k \in [n]}\{p_{n,k}\} \to 0$;*

*(c) $\sum_{k=1}^{n} \epsilon_{n,k} \to 0$.*

*Then, $S_n \xrightarrow{\mathcal{D}} \mathrm{Poisson}(\lambda)$ of a Poisson distribution with parameter $\lambda$ as $n \to \infty$.*

## G    Proofs in Section 2

*Proof of Proposition 1.* From the fact that $Y = U + \hat{f}(X)$, one can replace the $Y$ term in each probability with $U + \hat{f}(X)$, proving that the LHS and RHS probabilities are identical. $\qquad\square$

*Proof of Proposition 2.* By Lemma 6, we know that

$$Q_\tau(x) \in \arg\min_{u \in \mathbb{R}} \mathbb{E}\big[l_\tau(u, U)|X = x\big].$$

Since $l_\tau(u_1, u_2) = l_\tau(u_1 + c, u_2 + c)$, $\forall c \in \mathbb{R}$, we have

$$Q_\tau(x) \in \arg\min_{u \in \mathbb{R}} \mathbb{E}\big[l_\tau(u + \hat{f}(X), U + \hat{f}(X))|X = x\big],$$

which completes the proof. $\qquad\square$

## H    Proofs in Section 3.1

We start the analysis on a fixed point $x \in \mathcal{X}$. We first analyze the case where $n(x) > 0$, i.e. $\exists i \in [n], \|X_i - x\| \leq h$. We denote those $\|X_i - x\| \leq h$ by $\{X_{i_k}\}_{k=1}^{n(x)}$. We denote $F_{U|X_{i_k}}$ by $F_k$. We denote the average distribution of $F_k$ by $\bar{F} = \sum_{k=1}^{n(x)}\frac{1}{n(x)}F_k$, and denote its $\tau^\text{th}$ quantile by $Q_\tau(\bar{F})$. We abbreviate the estimation $\hat{Q}_\tau^{\mathrm{SNQ}}$ obtained by Algorithm 1 to be $\hat{Q}_\tau$ in this section. To bound the difference $\big|\hat{Q}_\tau(x) - Q_\tau(U|X = x)\big|$, we factorize it into two terms, as an analogy to the bias-variance factorization in nonparametric analysis:

$$\big|\hat{Q}_\tau(x) - Q_\tau(U|X = x)\big| \leq \big|Q_\tau(\bar{F}) - Q_\tau(U|X = x)\big| + \big|\hat{Q}_\tau(x) - Q_\tau(\bar{F})\big|,$$

where the former term is called the *bias* term and the latter the *variance* term. We now deal with them one by one.

**Lemma 4.** *Under Assumption 1 (a) with Lipschitz constant L, we have*

$$\left| Q_\tau(\bar{F}) - Q_\tau(U|X = x) \right| \leq Lh.$$

*Proof of Lemma 4.* By Assumption 1 (a), $\forall k \in [n(x)]$,

$$\left| Q_\tau(U|X = X_{i_k}) - Q_\tau(U|X = x) \right| \leq Lh,$$

which implies that

$$F_k\left( \left( Q_\tau(U|X = x) + Lh \right)^+ \right) = F_k\left( Q_\tau(U|X = x) + Lh \right) \geq \tau,$$

$$F_k\left( \left( Q_\tau(U|X = x) + Lh \right)^- \right) \leq \tau.$$

Hence,

$$\bar{F}\left( \left( Q_\tau(U|X = x) + Lh \right)^+ \right) = \bar{F}\left( Q_\tau(U|X = x) + Lh \right) = \sum_{k=1}^{n(x)} \frac{1}{n(x)} F_k\left( Q_\tau(U|X = x) + Lh \right) \geq \tau,$$

$$\bar{F}\left( \left( Q_\tau(U|X = x) + Lh \right)^- \right) = \sum_{k=1}^{n(x)} \frac{1}{n(x)} F_k\left( \left( Q_\tau(U|X = x) + Lh \right)^- \right) \leq \tau,$$

which justifies that

$$\left| Q_\tau(\bar{F}) - Q_\tau(U|X = x) \right| \leq Lh.$$

$\square$

Lemma 4 controls the bias term. For the variance term, we need a more careful analysis. We start with inspecting the properties of the expected pinball loss function. We will show that under Assumption 1 (b) and (c), the empirical risk minimization solution obtained at the second step of Algorithm 1 is actually close to the true quantile of $\bar{F}$ by two steps:

**Step 1.** The empirical risk minimizer $\hat{Q}_\tau(x)$ concentrates around its corresponding population risk's unique minimizer for sufficiently large $n(x) \geq C(\underline{p}, \underline{r}, M)$, where the minimizer is denoted by $u^*$.

**Step 2.** The unique population risk minimizer $u^*$ in Step 1 is identical to the population risk minimizer of $\mathbb{E}_{U \sim \bar{F}}[l_\tau(u, U)]$.

We prove the latter Step 2 first due to its simplicity. The following Lemma 5 proves Step 2.

**Lemma 5.** *We define the population risk stated in Step 1 by $\bar{l}_\tau(u) := \mathbb{E}_{U_k \sim F_k}[\sum_{k=1}^{n(x)} \frac{1}{n(x)} l_\tau(u, U_k)]$. Suppose Assumption 1 hold for some $L, \underline{p}, \underline{r}, M$. Then*

*(a) $\bar{l}_\tau$ is twicely differentiable and convex;*

*(b) $\bar{l}_\tau$ is identical to $\mathbb{E}_{U \sim \bar{F}}[l_\tau(u, U)]$ up to a constant;*

*(c) For sufficiently small $h \leq \frac{r}{2L}$, $\exists r_0 \geq \frac{r}{2}$, s.t. the minimizer of $\bar{l}_\tau$ is unique (denoted by $u^*$), $u^* \in [-M, M]$, and $\bar{l}_\tau$ is $\underline{p}$-strongly convex within a ball of radius $r_0$ around its minimizer $u^*$.*

To prove Lemma 5, we introduce the following Lemma 6, whose proof is postponed to Appendix J.

**Lemma 6.** *For any distribution with cdf $F(u) = \mathbb{P}(U \leq u)$, the expected pinball loss with respect to $F$ is semi-derivative with respect to $u$, and*

$$\partial_- \mathbb{E}_{U \sim F}[l_\tau(u, U)] = F(u^-) - \tau,$$
$$\partial_+ \mathbb{E}_{U \sim F}[l_\tau(u, U)] = F(u) - \tau.$$

*Furthermore, for those $F(u^-) = F(u)$, the derivative exists.*

*Proof of Lemma 5.* By Lemma 6, we have

$$\nabla_u \bar{l}_\tau(u) = \frac{1}{n(x)} \sum_{k=1}^{n(x)} \nabla_u \mathbb{E}_{U \sim F_k}[l_\tau(u, U)] = \frac{1}{n(x)} \sum_{k=1}^{n(x)} F_k(u) - \tau.$$

Since we assume that $(X, U)$ follows a continuous distribution, and the conditional pdf exists, then

$$\nabla^2 \bar{l}_\tau(u) = \frac{1}{n(x)} \sum_{k=1}^{n(x)} p_k(u),$$

where $p_k$ is the pdf of $F_k$, $\forall k \in [n(x)]$, which proves part (a).

For part (b), one only needs to notice that $\bar{F}$ has the cdf of the form $\sum_{k=1}^{n(x)} \frac{1}{n(x)} F_k$, which implies that $\mathbb{E}_{U \sim \bar{F}}[l_\tau(u, U)]$ has the same derivative as $\bar{l}_\tau$.

For part (c), by Assumption 1 (c), we have $n(x)$ balls $B(Q_\tau(U|X = X_{i_k}), \underline{r})$ $k \in [n(x)]$, such that

$$p_k(u) \geq \underline{p}, \quad \forall u \in B\big(Q_\tau(U|X = X_{i_k}), \underline{r}\big).$$

At the same time, from the proof of Lemma 4, we know that

$$\big|Q_\tau(\bar{F}) - Q_\tau(U|X = X_{i_k})\big| \leq Lh \leq \frac{1}{2} \cdot \underline{r},$$

which means that

$$
\begin{aligned}
\max_k \{Q_\tau(U|X = X_{i_k}) - \underline{r}\} + \frac{1}{2} \cdot \underline{r} &\leq \min_k Q_\tau(U|X = X_{i_k}) \\
&\leq Q_\tau(\bar{F}) \\
&\leq \max_k Q_\tau(U|X = X_{i_k}) \\
&\leq \min_k \{Q_\tau(U|X = X_{i_k}) + \underline{r}\} - \frac{1}{2} \cdot \underline{r}.
\end{aligned}
$$

Such an inequality implies that for at least a ball of radius $\frac{r}{2}$,

$$\nabla^2 \mathbb{E}_{U \sim \bar{F}}[l_\tau(u, U)] = \frac{1}{n(x)} \sum_{k=1}^{n(x)} p_k(u) \geq \underline{p}, \quad \forall u \in B(Q_\tau(\bar{F}), \frac{r}{2}),$$

which proves the uniqueness of $u^*$ and the local strong convexity.

Since $\min_k Q_\tau(U|X = X_{i_k}) \leq Q_\tau(\bar{F}) \leq \max_k Q_\tau(U|X = X_{i_k})$ and $Q_\tau(U|X = X_{i_k}) \in [-M, M]$, we have

$$u^* = Q_\tau(\bar{F}) \in [-M, M].$$

$\square$

Now we go back to prove Step 1. To get the optimal convergence rate, our arguments share the same spirits of Bartlett et al. [2005], Koltchinskii [2006] of getting fast convergence rates for the M-estimators of low variance. But for simplicity, we derive our proof of uniform convergence via Bernstein's inequality and covering number arguments, which is more similar to the way of Maurer and Pontil [2009].

We are now interested in

$$\xi_k(u) := l_\tau(u, U_{i_k}) - l_\tau(Q_\tau(\bar{F}), U_{i_k}), \quad \forall u \in [-M, M], \ U_{i_k} \sim U|X = X_{i_k}, \ k \in [n(x)].$$

A direct computation shows that

**Lemma 7.** $\forall u \in [-M, M]$, we have

$$|\xi_k(u)| \leq \max\{\tau, 1 - \tau\} \big|u - Q_\tau(\bar{F})\big| \leq \big|u - Q_\tau(\bar{F})\big|,$$

$$\mathrm{Var}(\xi_k(u)) \leq \frac{1}{3} \max\{\tau^2, (1 - \tau)^2\} \big|u - Q_\tau(\bar{F})\big|^2 \leq \frac{1}{3} \big|u - Q_\tau(\bar{F})\big|^2.$$

*Proof of Lemma 7.* Note that the pinball loss can be expressed in an alternative way:

$$l_\tau(u, U) = \max\{(1 - \tau)(u - U), \tau(U - u)\},$$

$$l_\tau(Q_\tau(\bar{F}), U) = \max\big\{(1 - \tau)\big(Q_\tau(\bar{F}) - U\big), \ \tau\big(U - Q_\tau(\bar{F})\big)\big\}.$$

From the fact that $\big|(1 - \tau)(u - U) - (1 - \tau)(Q_\tau(\bar{F}) - U)\big| \leq (1 - \tau)\big|u - Q_\tau(\bar{F})\big|$ and that $\big|\tau(U - u) - \tau\big(U - Q_\tau(\bar{F})\big)\big| \leq \tau|u - Q_\tau(\bar{F})|$, we prove the first statement via Lemma 15.

For the second claim, one just need to notice that any almost surely bounded by $[a, b]$ random variables have at most a variance of $\frac{1}{12}(b - a)^2$. $\square$

Combining Lemma 1 and Lemma 7, we directly prove the following

**Lemma 8** (Pointwise Convergence). $\forall \delta > 0, n(x) > 0, u \in [-M, M]$, *with probability at least* $1 - \delta$ *the following holds*

$$\left| \frac{1}{n(x)} \sum_{k=1}^{n(x)} \mathbb{E}[\xi_k(u)] - \frac{1}{n(x)} \sum_{k=1}^{n(x)} \xi_k(u) \right| \leq \frac{2}{3n(x)} \left| u - Q_\tau(\bar{F}) \right| \log\left(\frac{1}{\delta}\right)$$

$$+ \sqrt{\frac{2}{3n(x)} \left| u - Q_\tau(\bar{F}) \right|^2 \log\left(\frac{1}{\delta}\right)}.$$

*Proof.* Direct corollary from Lemma 1 and Lemma 7. $\qquad\square$

To reach a uniform convergence guarantee from pointwise convergence, we utilize the covering number arguments.

**Definition 4** ($\epsilon$-Covering Numbers). $\forall \epsilon > 0$, a function class $\mathcal{F}$ is said to have an $\epsilon$-covering number of

$$N\left(\epsilon, \mathcal{F}, \|\cdot\|_\infty\right) := \inf\{|\mathcal{F}_0| \colon \mathcal{F}_0 \subset \mathcal{F}, \forall f \in \mathcal{F}, \exists f_0 \in \mathcal{F}_0, \text{s.t.} \|f - f_0\|_\infty \leq \epsilon\}.$$

We compute the function class formed up by the functions $l_\tau(u, \cdot) - l_\tau(Q_\tau(\bar{F}), \cdot)$.

**Lemma 9** (Covering Number). *Assume Assumption 1 (b) holds with parameter* $M$. *Define* $\mathcal{L}_{[-M,M]} := \{l_\tau(u, \cdot) - l_\tau(Q_\tau(\bar{F}), \cdot) \colon u \in [-M, M]\}$. *Then* $\forall \epsilon > 0$,

$$N\left(\epsilon, \mathcal{L}_{[-M,M]}, \|\cdot\|_\infty\right) \leq \frac{\operatorname{diam}([-M, M])}{\epsilon} = \frac{2M}{\epsilon}.$$

*Proof of Lemma 9.* By a similar argument to that in Lemma 7, we have

$$\left| \left(l_\tau(u_1, \cdot) - l_\tau(Q_\tau(\bar{F}), \cdot)\right) - \left(l_\tau(u_2, \cdot) - l_\tau(Q_\tau(\bar{F}), \cdot)\right) \right| \leq |u_1 - u_2|,$$

where the left-hand-side is the function infinity form by itself.

Combining the above fact with another that

$$|u_1 - u_2| \leq \sup_{u_1', u_2' \in [-M, M]} |u_1' - u_2'| = \operatorname{diam}([-M, M]),$$

we complete the proof. $\qquad\square$

Since there will be many coefficients dependence, we abbreviate any constant $C$ that depends polynomially on some other factors $(c_1, \cdots, c_q)$ by $C(c_1, \cdots, c_q)$. Now we provide the uniform convergence result by combining the pointwise convergence and the covering number.

**Lemma 10** (Uniform Convergence). $\forall \epsilon, \delta > 0$, *assume* $n(x)$ *is sufficiently large such that* $n(x) \geq \frac{2}{3} \log(\frac{2M}{\epsilon\delta})$, *we have with probability at least* $1 - \delta$,

$$\left| \frac{1}{n(x)} \sum_{k=1}^{n(x)} \mathbb{E}[\xi_k(u)] - \frac{1}{n(x)} \sum_{k=1}^{n(x)} \xi_k(u) \right| \leq 4\epsilon + \frac{2}{3n(x)} \left| u - Q_\tau(\bar{F}) \right| \log\left(\frac{2M}{\epsilon\delta}\right)$$

$$+ \sqrt{\frac{2}{3n(x)} \left| u - Q_\tau(\bar{F}) \right|^2 \log\left(\frac{2M}{\epsilon\delta}\right)}$$

*holds for* $\forall u \in [-M, M]$.

*Specifically, if we select* $\epsilon = \frac{1}{n(x)}, \delta = \frac{1}{n(x)M}$,
*we have* $\forall n(x) \geq C_1(\log(M)) := 6 \log(\max\{2M, 10\})$,

$$\left| \frac{1}{n(x)} \sum_{k=1}^{n(x)} \mathbb{E}[\xi_k(u)] - \frac{1}{n(x)} \sum_{k=1}^{n(x)} \xi_k(u) \right| \leq \frac{1}{n(x)} \left( 4 + \frac{8M}{3} \log(2n(x)M) \right)$$

$$+ \sqrt{\frac{4}{3n(x)} \left| u - Q_\tau(\bar{F}) \right|^2 \log(2n(x)M)}$$

*holds for* $\forall u \in [-M, M]$.

*Proof of Lemma 10.* $\forall \epsilon > 0$, we can select an $\epsilon$-cover of the function class $\mathcal{L}_{[-M,M]}$, denoted by

$$\mathcal{S} = \{L[u_j] = l_\tau(u_j, \cdot) - l_\tau(Q_\tau(\bar{F}), \cdot)\}_{j=1}^{N(\epsilon, \mathcal{L}_{[-M,M]}, \|\cdot\|_\infty)}.$$

Note that

$$\xi_k(u) = l_\tau(u, U_{i_k}) - l_\tau(Q_\tau(\bar{F}), U_{i_k}) = L[u](U_{i_k}), \quad \forall u \in [-M, M],$$

which means that $\exists u_j \in \mathcal{S}$, s.t.

$$|\xi_k(u) - \xi_k(u_j)| = |L[u](U_{i_k}) - L[u_j](U_{i_k})| \le \|L[u] - L[u_j]\|_\infty \le \epsilon.$$

Now $\forall \delta > 0$, we apply Lemma 8 on $\{\xi_k(u_j)\}_{k=1}^m$, $u_j \in \mathcal{S}$ with probability guarantee of at least $1 - \frac{\delta}{N(\epsilon, \mathcal{L}_{[-M,M]}, \|\cdot\|_\infty)}$, we have $\forall u_j \in \mathcal{S}$,

$$\left| \frac{1}{m} \sum_{k=1}^m \mathbb{E}[\xi_k(u_j)] - \frac{1}{m} \sum_{k=1}^m \xi_k(u_j) \right| \le \frac{2}{3m} |u_j - Q_\tau(\bar{F})| \log \left( \frac{N(\epsilon, \mathcal{L}_{[-M,M]}, \|\cdot\|_\infty)}{\delta} \right)$$
$$+ \sqrt{\frac{2}{3m} |u_j - Q_\tau(\bar{F})|^2 \log \left( \frac{N(\epsilon, \mathcal{L}_{[-M,M]}, \|\cdot\|_\infty)}{\delta} \right)}.$$

Combining above inequality with the fact that $|\xi_k(u) - \xi_k(u_j)| \le \epsilon$ and the way we select the cover in Lemma 9 such that $|u - u_j| \le \epsilon$, we complete the proof.

To verify the specific conclusion, one just need to notice that our choice of $n(x)$ ensures that $n(x) \ge \frac{4}{3} \log(2n(x)M)$ for $\epsilon = \frac{1}{n(x)}, \delta = \frac{1}{n(x)M}$. $\qquad\square$

**Lemma 11** (Generalization Bound for $\hat{Q}_\tau$). *If we take $\hat{Q}_\tau(x) = \arg\min_u \sum_{k=1}^{n(x)} \frac{1}{n(x)} l_\tau(u, U_{i_k})$ as we do in Algorithm 1, then for sufficiently large $m \ge C_2(\underline{p}^{-1}, \underline{r}^{-1}, M)$, we have with probability at least $1 - \frac{1}{n(x)M}$,*

$$0 \le \bar{l}_\tau(\hat{Q}_\tau(x)) - \bar{l}_\tau(Q_\tau(\bar{F})) \le \frac{C_3(\underline{p}^{-1}, M, \log(n(x)M))}{n(x)} = \tilde{O}\left(\frac{1}{n(x)}\right),$$

*where $C_3 := 8 + \frac{8M}{3} \log(n(x)M) + \frac{16 \log(n(x)M)}{3\underline{p}}$.*

*More specifically, with probability at least $1 - \frac{1}{n(x)M}$,*

$$\left| \hat{Q}_\tau(x) - Q_\tau(\bar{F}) \right| \le \frac{C_4(\underline{p}^{-1}, M, \log(n(x)M))}{\sqrt{n(x)}} = \tilde{O}\left(\frac{1}{\sqrt{n(x)}}\right),$$

*where $C_4 := \sqrt{C_3 \cdot \frac{2}{\underline{p}}}$.*

*Proof of Lemma 11.* We first claim that for sufficiently large $n(x) \ge C_2$, the empirical minimizer $\hat{Q}_\tau(x)$ must fall into the neighborhood $B(Q_\tau(\hat{F}), \frac{\underline{r}}{2})$ stated in Lemma 5 (c). In fact, one just needs to verify that

$$\frac{1}{n(x)} \left( 4 + \frac{8M}{3} \log(2n(x)M) \right) + \frac{1}{\sqrt{n(x)}} \cdot \sqrt{\frac{16}{3} M^2 \log(2n(x)M)} \le \frac{\underline{p} \cdot \underline{r}^2}{8},$$

which will definitely hold for sufficiently large $n(x) \ge C_2(\underline{p}^{-1}, \underline{r}^{-1}, M)$.

From the fact that minimizing $\sum_{k=1}^{n(x)} \frac{1}{n(x)} l_\tau(u, U_{i_k})$ is equivalent to minimizing it minus some constant, we know that

$$\sum_{k=1}^{n(x)} \frac{1}{n(x)} \xi_k(\hat{Q}_\tau(x)) \le 0.$$

By Lemma 5 and Lemma 10, we have

$$0 \le \bar{l}_\tau(\hat{Q}_\tau(x)) - \bar{l}_\tau(Q_\tau(\bar{F}))$$

$$= \frac{1}{n(x)} \sum_{k=1}^{n(x)} \mathbb{E}[\xi_k(\hat{Q}_\tau(x)]$$

$$\le \frac{1}{n(x)} \left( 4 + \frac{8M}{3} \log(2n(x)M) \right) + \sqrt{\frac{16}{3n(x)} |\hat{Q}_\tau(x) - Q_\tau(\bar{F})|^2 \log(2n(x)M)}$$

$$\le \frac{1}{n(x)} \left( 4 + \frac{8M}{3} \log(2n(x)M) \right) + \sqrt{\frac{16}{3n(x)} \cdot \frac{2}{\underline{p}} \left( \bar{l}_\tau(\hat{Q}_\tau(x)) - \bar{l}_\tau(Q_\tau(\bar{F})) \right) \log(2n(x)M)},$$

where the first inequality follows from Lemma 5 (a), the second inequality from Lemma 10, and the last from Lemma 5 (c).

Solve the quadratic inequality

$$w^2 \le \frac{\alpha}{n(x)} + \sqrt{\frac{\beta}{n(x)}} w,$$

where $\alpha = 4 + \frac{8M}{3} \log(2n(x)M), \beta = \frac{32 \log(2n(x)M)}{3\underline{p}}$, we have

$$w \le \frac{\sqrt{\frac{\beta}{n(x)}} + \sqrt{\frac{\beta}{n(x)} + \frac{4\alpha}{n(x)}}}{2} \le \frac{1}{\sqrt{n(x)}} \left( \sqrt{\beta} + \sqrt{\alpha} \right),$$

where $w$ is exactly $\sqrt{\bar{l}_\tau(\hat{Q}_\tau(x)) - \bar{l}_\tau(Q_\tau(\bar{F}))}$.

Hence

$$\bar{l}_\tau(\hat{Q}_\tau(x)) - \bar{l}_\tau(Q_\tau(\bar{F})) \le \frac{1}{n(x)} (2\beta + 2\alpha) = \frac{1}{n(x)} \left( 8 + \frac{16M}{3} \log(2n(x)M) + \frac{64 \log(2n(x)M)}{3\underline{p}} \right).$$

The latter conclusion follows from the local strong convexity of $\bar{l}_\tau$ again, as is shown in Lemma 5 (c). $\qquad\square$

Lemma 11 proves our Step 1, finally. We can combine Step 1 and Step 2 now to get an upper bound that is critical for the following analysis:

**Lemma 12** (Bounding $\left| \hat{Q}_\tau(x) - Q_\tau(U|X = x) \right|$). *Suppose Assumption 1 holds. For sufficiently small $h \le \frac{r}{2L}$, we have*

$$\left| \hat{Q}_\tau(x) - Q_\tau(U|X = x) \right| \le Lh + \mathbb{1}\{n(x) < C_2\}M$$

$$+ \mathbb{1}\{n(x) \ge C_2\} \left( M\mathbb{1}\{\text{Bounds in Lemma 11 fail}\} + \frac{C_4}{\sqrt{n(x)}} \right),$$

*where $C_2 = C_2(\underline{p}^{-1}, \underline{r}^{-1}, M)$, $C_4 = C_4(\underline{p}^{-1}, M, \log(n(x)M))$ in Lemma 11, and the failing probability in Lemma 11 is no larger than $\frac{1}{n(x)M}$ provided $n(x) \ge C_2$.*

*Proof of Lemma 12.* A direct corollary from Lemma 4 and Lemma 11. $\qquad\square$

*Proof of the pointwise consistency part of Theorem 1.* Since the $\tilde{O}\left( \frac{1}{\sqrt{n(x)}} \right)$ term and the failing probability term in Lemma 12 tend to zero as $n(x) \to \infty$, we only need to verify that we can select $h$ such that as $n \to \infty$, we have

(a) $h \to 0$;

(b) $\forall C > 0, \mathbb{P}(n(x) \geq C) \to 1$.

We will show that both conditions are true for any $x_0$ where the pdf $p(x)$ is positive and continuous at $x = x_0$, if we select $h = C_5 n^{-\frac{1}{d+2}}$.

The first argument automatically holds as $n \to \infty$. We turn to prove the second one.

Since $p(x)$ is continuous at $p(x_0)$, we can find sufficiently large $n \geq N_0$, such that

$$p(x) \geq \frac{p(x_0)}{2}, \quad \forall x \in B(x_0, h).$$

Then the probability of $\mathbb{1}\{X_i \in B(x_0, h)\}$ can be lower bounded by

$$\mathbb{P}(X_i \in B(x_0, h)) \geq \frac{p(x_0)}{2} V h^d = C_6 n^{-\frac{d}{d+2}},$$

where $V$ is the volume of the unit ball in $\mathcal{X} \subset \mathbb{R}^d$, $C_6 := \frac{p(x_0)}{2} V C_5$.

$\forall C > 0, j \in \mathbb{N}_+$, we can define a sequence of parameters $\{\lambda_j\}_{j=1}^{\infty}$, such that for any $\eta_j \sim$ Poisson$(\lambda_j)$,

$$\mathbb{P}(\eta_j \geq C) \geq 1 - \frac{1}{j}, \quad j = 1, 2, \ldots,$$

where we assume without loss of generality that $\lambda_j$ is non-decreasing.

For each $j$, we can define an auxiliary sequence of independent random variables $\{\nu_{l,q,j}\}$ for $l = [q], q = 1, 2, \ldots, j = 1, 2, \ldots$, such that each random variable $\nu_{l,q,j} \sim$ Bernoulli$(\frac{\lambda_j}{q})$.

By Lemma 3, we can find a sequence of $\{q_j\}_{j=1}^{\infty}$, such that

$$\mathbb{P}\left(\sum_{l=1}^{q} \nu_{l,q,j} \geq C\right) \geq \mathbb{P}(\eta_j \geq C) - \frac{1}{j} \geq 1 - \frac{2}{j}, \quad \forall q \geq q_j, j = 1, 2, \ldots,$$

where we select $q_j$ without loss of generality that $\frac{\lambda_j}{q_j}$ is non-increasing.

As $\{\mathbb{1}\{X_i \in B(x_0, h)\}\}_{i=1}^{n}$ is also an independent Bernoulli random variable sequence with parameter $C_6 n^{-\frac{d}{d+2}}$, we can select a non-decreasing sequence of $\{N_j\}_{j=1}^{\infty}$, such that $\forall n \geq N_j$,

$$C_6 n^{-\frac{d}{d+2}} \geq \frac{\lambda_j}{n}, \quad n \geq q_j.$$

Then $\forall n \geq N_j$, we have

$$\mathbb{P}\left(n(x_0) = \sum_{i=1}^{n} \mathbb{1}\{X_i \in B(x_0, h)\} \geq C\right) \geq \mathbb{P}\left(\sum_{i=1}^{n} \nu_{i,n,j} \geq C\right) \geq 1 - \frac{2}{j},$$

where the first inequality follows from the result of stochastically dominant of $\mathbb{1}\{X_i \in B(x_0, h)\}$ over $\nu_{l,n,j}$.

$\forall \epsilon > 0$, we can select sufficiently large $j \geq \frac{2}{\epsilon}$, then $\forall n \geq \max\{N_0, N_j\}$,

$$\mathbb{P}(n(x_0) \geq C) \geq 1 - \epsilon,$$

which completes the proof. $\qquad\square$

To prove the mean squared error part of Theorem 1, we state a lemma on the Binomial random variable that will be proved later.

**Lemma 13.** *For any $\zeta \sim$ Binomial$(n, p)$, we have*

*(a)* $\mathbb{E}\left[\frac{1}{\zeta}\mathbb{1}\{\zeta > 0\}\right] \leq \frac{2}{(n+1)p}$;

*(b)* $\forall r \in \mathbb{N}, r < np$, we have $\mathbb{P}(\zeta \leq r) \leq \frac{(n-r)p}{(np-r)^2}$.

Now we return to the proof of the mean squared error part of Theorem 1.

*Proof of the mean squared error part of Theorem 1.* By direct computation from Lemma 12, we have

$$
\mathbb{E}\left[|\hat{Q}_\tau(X_{\text{test}}) - Q_\tau(U|X = X_{\text{test}})|^2\right]
$$

$$
= \mathbb{E}_{X_{\text{test}}}\left[\mathbb{E}_{X_{1:n}}\left[\mathbb{E}_{U_{1:n}}[|\hat{Q}_\tau(X_{\text{test}}) - Q_\tau(U|X = X_{\text{test}})|^2]\right]\right]
$$

$$
\leq \mathbb{E}_{X_{\text{test}}}\Bigg[\mathbb{E}_{X_{1:n}}\bigg[4L^2h^2
$$

$$
+ \Big[\frac{4C_4(\underline{p}^{-1}, \underline{M}, \log(n(X_{\text{test}})M))^2}{n(X_{\text{test}})} + 4M^2(\frac{1}{n(X_{\text{test}})M})^2\Big]\mathbb{1}\{n(X_{\text{test}}) \geq C_2\}
$$

$$
+ 4M^2\mathbb{1}\{n(X_{\text{test}}) < C_2\}\bigg]\Bigg]
$$

$$
\leq 4L^2h^2 + \mathbb{E}_{X_{\text{test}}}\left[\mathbb{E}_{X_{1:n}}\left[\frac{C_4'(\underline{p}^{-1}, \underline{M}, \log(nM))}{n(X_{\text{test}})}\mathbb{1}\{n(X_{\text{test}}) \geq C_2\} + 4M^2\mathbb{1}\{n(X_{\text{test}}) < C_2\}\right]\right],
$$

where the first inequality comes from the fact that $(a + b + c + d)^2 \leq 4(a^2 + b^2 + c^2 + d^2)$, and the second from that $n(X_{\text{test}}) \leq n$. Note that we replace the original $C_4$ that depends on the logarithmic term of $n(X_{\text{test}})$ with a new $C_4'$ that does not depend logarithmically on $n(X_{\text{test}})$ but logarithmically on the entire $n$. From the definition of $C_4$ in Lemma 11, we recall that the dependence on the logarithmic term is polynomial.

By Lemma 13 (a), we have

$$
\mathbb{E}_{X_{1:n}}\left[\frac{1}{n(X_{\text{test}})}\mathbb{1}\{n(X_{\text{test}} \geq C_2)\}\right] \leq \mathbb{E}_{X_{1:n}}\left[\frac{1}{n(X_{\text{test}})}\mathbb{1}\{n(X_{\text{test}} > 0)\}\right]
$$

$$
\leq \frac{2}{(n+1)\mathbb{P}(X_i \in B(X_{\text{test}}, h))}.
$$

We try to bound $\mathbb{E}_X[\frac{1}{\mathbb{P}(X_i \in B(X, h))}]$.

Since $\mathcal{X} \subset [0, 1]^d$, we can generate an $\frac{h}{2}$-cover of $\mathcal{X}$, denoted by $\mathcal{S} = \{x_j\}_{j=1}^{C_d}$, where

$$
C_d \leq \frac{C}{h^d},
$$

where $C$ is some intrinsic constant with respect to the space $\mathbb{R}^d$ equipped with metric $\|\cdot\|$.

Then

$$
\mathbb{E}_X\left[\frac{1}{\mathbb{P}(X_i \in B(X, h))}\right] = \int_{\mathcal{X}}\frac{1}{P(B(x, h))}\mathrm{d}P
$$

$$
\leq \sum_{j=1}^{C_d}\int_{\mathcal{X}}\frac{\mathbb{1}\{x \in B(x_j, \frac{h}{2})\}}{P(B(x, h))}\mathrm{d}P
$$

$$
\leq \sum_{j=1}^{C_d}\int_{\mathcal{X}}\frac{\mathbb{1}\{x \in B(x_j, \frac{h}{2})\}}{P(B(x_j, \frac{h}{2}))}\mathrm{d}P
$$

$$
= C_d \leq \frac{C}{h^d},
$$

where $P$ is the probability measure with respect to $X$, and the first inequality follows from that $\mathcal{S}$ is a $\frac{h}{2}$-cover of $\mathcal{X}$, the second from the fact that $B(x_j, \frac{h}{2}) \subset B(x, h)$ for those $\|x - x_j\| \leq \frac{h}{2}$, and the inequality from the fact that $\int \mathbb{1}\{x \in B(x_j, \frac{h}{2})\}\mathrm{d}P = P(B(x_j, \frac{h}{2}))$.

We now try to bound $\mathbb{E}_{X_{\text{test}}}\left[\mathbb{E}_{X_{1:n}}\left[\mathbb{1}\{n(X_{\text{test}}) < C_2\}\right]\right]$.

If $nP(B(x,h)) \geq 2\lfloor C_2 \rfloor$, then by Lemma 13 (b), we have

$$\mathbb{P}(n(x) < C_2) \leq \frac{n\lfloor C_2 \rfloor}{(nP(B(x,h)) - \lfloor C_2 \rfloor)^2} \leq \frac{4}{nP(B(x,h))}.$$

If $nP(B(x,h)) < 2\lfloor C_2 \rfloor$, then for sufficiently large $n \geq 4\lfloor C_2 \rfloor - 1$, by direct calculation we have,

$$\frac{P(B(x,h))}{1 - P(B(x,h))} \leq 2\lfloor C_2 \rfloor \cdot \frac{2}{n-1} \leq \frac{4}{3}\lfloor C_2 \rfloor \cdot \frac{3}{n-2} \leq \lfloor C_2 \rfloor \cdot \frac{4}{n-3} \leq \cdots \leq \lfloor C_2 \rfloor \cdot \frac{\lfloor C_2 \rfloor}{n+1-\lfloor C_2 \rfloor},$$

which implies that

$$\mathrm{C}_n^l \left(1 - P(B(x,h))\right)^{n-l} \left(P(B(x,h))\right)^l \leq 2 \cdot \frac{4}{3} \cdot \lfloor C_2 \rfloor \cdot \mathrm{C}_n^0 \left(1 - P(B(x,h))\right)^n, \quad \forall l \in [\lfloor C_2 \rfloor],$$

where $\mathrm{C}_n^l$ is the number of combinations of $(n, l)$. Hence

$$\begin{aligned}
\mathbb{P}(n(x) &< C_2) \\
&\leq \mathbb{P}(n(x) \leq \lfloor C_2 \rfloor) \\
&= \sum_{l=0}^{\lfloor C_2 \rfloor} \mathrm{C}_n^l \left(1 - P(B(x,h))\right)^{n-l} \left(P(B(x,h))\right)^l \\
&\leq \frac{8(C_2)^2}{3} \left(1 - P(B(x,h))\right)^n \\
&\leq \frac{8(C_2)^2}{3} \exp(-nP(B(x,h))).
\end{aligned}$$

Since $\exp(-a) \leq \frac{1}{a} \cdot \max_b\{b\exp(-b)\} = \frac{1}{ea}$, we have

$$\mathbb{P}(n(x) < C_2) \leq \frac{C_7}{nP(B(x,h))},$$

where $C_7 = \frac{8(C_2)^2}{3e}$.

Therefore, we can conclude that (w.l.o.g. we assume $C_7 \geq 4$)

$$\mathbb{P}(n(x) < C_2) \leq \frac{\max\{C_7, 4\}}{nP(B(x,h))} = \frac{C_7}{nP(B(x,h))}.$$

Hence, we can finally give the bound by

$$\begin{aligned}
\mathbb{E}&\left[|\hat{Q}_\tau(X_{\text{test}}) - Q_\tau(U|X = X_{\text{test}})|^2\right] \\
&\leq 4L^2h^2 + \mathbb{E}_{X_{\text{test}}}\left[\mathbb{E}_{X_{1:n}}\left[\frac{C_4'(\underline{p}^{-1}, \underline{M}, \log(nM))}{n(X_{\text{test}})}\mathbb{1}\{n(X_{\text{test}}) \geq C_2\} + 4M^2\mathbb{1}\{n(X_{\text{test}}) < C_2\}\right]\right] \\
&\leq 4L^2h^2 + C_4'Ch^{-d} \cdot \frac{1}{n} + C_7Ch^{-d} \cdot \frac{1}{n} \\
&= 4L^2h^2 + C_2'(\underline{p}^{-1}, \underline{r}^{-1}, M, \log(nM))h^{-d} \cdot \frac{1}{n}.
\end{aligned}$$

For the special case of $L = 0$, we can select $h = \Theta(M)$, hence reaching a fast rate of convergence of order $O\left(\frac{C_2'}{n}\right)$.

If $L > 0$, by selecting $h = n^{-\frac{1}{d+2}} L^{\frac{2}{d+2}} (dC_2')^{-\frac{1}{d+2}}$, we get the convergence result that

$$\mathbb{E}\left[|\hat{Q}_\tau(F_X) - Q_\tau(F_X)|^2\right] \leq L^{\frac{2d}{d+2}} \cdot n^{-\frac{2}{d+2}} \left(dC_2'(L, \underline{p}^{-1}, \underline{r}^{-1}, M, \log(n))\right)^{\frac{2}{d+2}},$$

where the dependence of $C_2'$ on $\log(n)$ is polynomial. We conclude that

$$\mathbb{E}\left[|\hat{Q}_\tau(F_X) - Q_\tau(F_X)|^2\right] = \tilde{O}\left(L^{\frac{2d}{d+2}} n^{-\frac{2}{d+2}}\right).$$

$\square$

*Remark* 2. A for large $d$ cases, the left behind term $(dC_2')^{\frac{2}{d+2}}$ is almost 1, compared to other terms. The major contribution to the error upper bound is made by the $L^{\frac{2d}{d+2}}n^{-\frac{2}{d+2}}$ term. If $L$ is replaced by $kL$, then $n$ must be at least $k^d n$ to keep the upper bound to remain the same, which implies that the contribution of $L$ is much more significant than $n$.

As for the lower bound, Theorem 2 follows immediately from Theorem 3.2 in Györfi et al. [2002] with Hölderness parameter $(1, L)$. Note that the class $\mathcal{P}^L$ we construct has a conditional distribution identical to the Gaussian of the unit variance, which automatically fulfills our Assumption 1 (b) and (c). The only additional requirement in $\mathcal{P}^L$ compared to their $\mathcal{D}^{(1,L)}$ is that we require the existence of $\mu(x) = 0$ for some $x \in [0,1]^d$. In their proof, they construct a sub-class based on the division of $[0,1]^d$ into smaller cubes, where the mean function $\mu(x)$ is $(1, L)$-Hölder (i.e. $L$-Lipschitz) on each cube while being zero on the boundary of those cubes. Hence our additional requirement of $\exists x \in [0,1]^d$ such that $\mu(x) = 0$ does not affect the result.

**Lemma 14** (Theorem 3.2 in Györfi et al. [2002])**.** *Consider a conditional mean estimation problem (i.e. regression problem) with the estimator sequence $\{\hat{\mu}_n\}$ and true conditional mean $\mu$. For the class $\mathcal{P}^L$, the sequence*

$$a_n = L^{\frac{2d}{d+2}}n^{-\frac{2}{d+2}}$$

*is a lower minimax rate of convergence. In particular,*

$$\liminf_{n \to \infty} \inf_{\hat{\mu}_n} \sup_{(X,Y) \sim \mathcal{P}, \mathcal{P} \in \mathcal{P}^L} \frac{\mathbb{E}\big[\|\hat{\mu}_n - \mu\|^2\big]}{L^{\frac{2d}{d+2}}n^{-\frac{2}{d+2}}} \geq C_8 > 0,$$

*for some constant $C_8$ independent of $L$.*

*Proof of Theorem 2.* Since for any Gaussian distribution with known variance, estimating its mean is equivalent to estimating its any $\tau$-th quantile, Theorem 2 is a direct corollary from Lemma 14, once one verifies that $\mathcal{P}^L$ satisfies Assumption 1.

Verifying Assumption 1 (a): Since the $\tau$-th conditional quantile is of the type $\mu(x) + \Phi(\tau)$ and $\mu(X)$ is $L$-Lipschitz, the conditional quantile is also of course $L$-Lipschitz.

Verifying Assumption 1 (b): Because $\mu(x)$ is $L$-Lipschitz and $\mathcal{X}$ is bounded, its image $\mu(\mathcal{X})$ is bounded. Furthermore, as we know $0 \in \mu(\mathcal{X})$, we can bound $\mu(\mathcal{X})$ by $[-L\sqrt{d}, L\sqrt{d}]$, leading to the boundedness of the conditional quantile set $\mu(\mathcal{X}) + \Phi(r)$.

Verifying Assumption 1 (c): The pdf around $\mu(x) + \Phi(\tau)$ is of the same shape as that of standard Gaussian at $\Phi(\tau)$, which is definitely lower bounded from zero in a neighborhood. $\square$

# I   Proofs in Section 3.2

*Proof of Theorem 3.* By the definition of mutual independence, we have $\forall A \in \mathcal{F}(A), B \in \mathcal{F}(U)$,

$$\mathbb{P}(U \in B | Z \in z(A))\mathbb{P}(X \in A | Z \in z(A)) = \mathbb{P}(U \in B, X \in A | Z \in z(A)).$$

Hence,

$$
\begin{aligned}
\mathbb{P}(U \in B | Z \in z(A)) &= \frac{\mathbb{P}(U \in B, X \in A | Z \in z(A))}{\mathbb{P}(X \in A | Z \in z(A))} \\
&= \frac{\mathbb{P}(U \in B, X \in A, Z \in z(A)) / \mathbb{P}(Z \in z(A))}{\mathbb{P}(X \in A, Z \in z(A)) / \mathbb{P}(Z \in z(A))} \\
&= \frac{\mathbb{P}(U \in B, X \in A, Z \in z(A))}{\mathbb{P}(X \in A, Z \in z(A))} \\
&= \frac{\mathbb{P}(U \in B, X \in A)}{\mathbb{P}(X \in A)} \\
&= \mathbb{P}(U \in B | X \in A),
\end{aligned}
$$

leading to the conclusion that

$$F_{U|X=x} = F_{U|Z=z(x)}.$$

Thus, if we have any consistency guarantee or mean squared error guarantee on
$$|\hat{Q}_\tau(U|Z = z(x)) - Q_\tau(U|Z = z(x))|,$$
then we also get the same result for
$$|\hat{Q}_\tau(U|Z = z(x)) - Q_\tau(U|X = x)|.$$
Notice that the dimension of $\mathcal{Z}$ is $d_0 \leq d$, we can conclude the proof. □

*Proof of Theorem 4.* By definition of quantiles, we know
$$\mathbb{P}\big(U \leq Q_\tau(U|Z) \,\big|\, Z\big) = F_{U|Z}(Q_\tau(U|Z)) \geq \tau.$$
Since we assume that $F_{U|Z}$ is absolutely continuous with respect to the Lebesgue measure, the Radon-Nikodym derivative exists, which is the conditional probability density function. Thus $F_{U|Z}$ must be locally Lipschitz. Then for any fixed bounded neighborhood $I$ containing $Q_\tau(U|Z)$, $F_{U|Z}$ is Lipschitz on $\bar{I}$ with some Lipschitz constant $L(\bar{I})$. Therefore $\forall \epsilon > 0$ and small enough such that $B(Q_\tau(U|Z), \epsilon) \subset I$,
$$F_{U|Z}(Q_\tau(U|Z) - \epsilon) \geq F_{U|Z}(Q_\tau(U|Z)) - L(I)\epsilon.$$
By the definition of quantiles, we have
$$F_{U|Z}(Q_\tau(U|Z) - \epsilon) < \tau.$$
Hence
$$F_{U|Z}(Q_\tau(U|Z)) < L(I)\epsilon + \tau.$$
Taking $\epsilon \to 0$, we have
$$F_{U|Z}(Q_\tau(U|Z)) \leq \tau.$$
Combining it with the fact that $F_{U|Z}(Q_\tau(U|Z)) \geq \tau$, we prove that
$$\mathbb{P}\big(U \leq Q_\tau(U|Z) \,\big|\, Z\big) = F_{U|Z}(Q_\tau(U|Z)) = \tau.$$
□

*Proof of Theorem 5.* We only prove the middle inequality, since the remaining two hold from similar arguments.

By a similar argument as Proposition 2, if one knows $Z^{(d_2)}$, then the best single point decision they can make with respect to conditional expectation of pinball loss is
$$Q_\tau(U|Z^{(d_2)}) \in \underset{u \in \mathbb{R}}{\arg\min}\, \mathbb{E}[l_\tau(u, U)|Z^{(d_2)}].$$
Since $Z^{(d_1)} \in \mathcal{F}(Z^{(d_2)})$, if another learner makes a decision based on the first $d_1$ components of $Z^{(d_2)}$, say, $Q_\tau(U|Z^{(d_1)})$, then they must suffer a pinball loss no smaller than that $Z^{(d_2)}$ learner, which is
$$\mathbb{E}[l_\tau(Q_\tau(U|Z^{(d_2)}), U)|Z^{(d_2)}] \leq \mathbb{E}[l_\tau(Q_\tau(U|Z^{(d_1)}), U)|Z^{(d_2)}].$$
By taking the expectation with respect to $Z^{(d_2)}$, the tower property of the conditional expectation tells us that
$$\begin{aligned}
\mathbb{E}\Big[l_\tau(Q_\tau(U|Z^{(d_2)}), U)\Big] &= \mathbb{E}\Big[\mathbb{E}[l_\tau(Q_\tau(U|Z^{(d_2)}), U)|Z^{(d_2)}]\Big] \\
&\leq \mathbb{E}\Big[\mathbb{E}[l_\tau(Q_\tau(U|Z^{(d_1)}), U)|Z^{(d_2)}]\Big] \\
&= \mathbb{E}\Big[l_\tau(Q_\tau(U|Z^{(d_1)}), U)\Big].
\end{aligned}$$
Noticing that $Y = U + \hat{f}(X)$ and $l_\tau(a + c, b + c) = l_\tau(a, b)$, we have
$$l_\tau(u + \hat{f}(X), Y) = l_\tau(u, U),$$
which verifies the proof finally. □

*Remark* 3. One may wonder if one can conduct the proof of Theorem 5 in a simpler way via Jensen's inequality since $l_\tau(\cdot, \cdot)$ is convex with respect to its first component. But unfortunately, such an argument does not work for the quantile case, since in general,
$$\mathbb{E}[Q_\tau(U|X)] \neq Q_\tau(U).$$

# J   Proofs of Auxiliary Lemmas in Appendix H

*Proof of Lemma 6.* Denote the probability measurement by $P$. From the definition of pinball loss and expectation, we have

$$\mathbb{E}_U[l_\tau(u, U)] = \int_{-\infty}^{u^-} (\tau - 1)(U - u)\mathrm{d}P + \int_{u^-}^{\infty} \tau(U - u)\mathrm{d}P$$

$$= \int_{-\infty}^{u} (\tau - 1)(U - u)\mathrm{d}P + \int_{u}^{\infty} \tau(U - u)\mathrm{d}P$$

For a small but positive $\Delta > 0$, we have

$$\mathbb{E}_U[l_\tau(u - \Delta, U)] - \mathbb{E}_U[l_\tau(u, U)]$$

$$= \int_{-\infty}^{u^- - \Delta} (\tau - 1)(U - u + \Delta)\mathrm{d}P + \int_{u^- - \Delta}^{\infty} \tau(U - u + \Delta)\mathrm{d}P$$

$$- \int_{-\infty}^{u^-} (\tau - 1)(U - u)\mathrm{d}P - \int_{u^-}^{\infty} \tau(U - u)\mathrm{d}P$$

$$= \int_{-\infty}^{u^-} (\tau - 1)(U - u + \Delta)\mathrm{d}P + \int_{u^-}^{\infty} \tau(U - u + \Delta)\mathrm{d}P$$

$$- \int_{-\infty}^{u^-} (\tau - 1)(U - u)\mathrm{d}P - \int_{u^-}^{\infty} \tau(U - u)\mathrm{d}P$$

$$- \int_{u^- - \Delta}^{u^-} (\tau - 1)(U - u + \Delta)\mathrm{d}P + \int_{u^- - \Delta}^{u^-} \tau(U - u + \Delta)\mathrm{d}P$$

$$= \Delta \left[ \int_{-\infty}^{u^-} (\tau - 1)\mathrm{d}P + \int_{u^-}^{\infty} \tau\mathrm{d}P \right]$$

$$+ \int_{u^- - \Delta}^{u^-} (U - u + \Delta)\mathrm{d}P$$

$$= \Delta(\tau - F(u^-)) + \int_{u^- - \Delta}^{u^-} (U - u + \Delta)\mathrm{d}P.$$

Note that any cdf is càdlàg (right continuous with left limits). Thus, $\forall \epsilon > 0, \exists \delta > 0$, s.t. $\forall \Delta < \delta$,

$$|F(u^- - \Delta) - F(u^-)| \leq \epsilon.$$

Then we have

$$0 \leq \int_{u^- - \Delta}^{u^-} (U - u + \Delta)\mathrm{d}P$$

$$\leq \int_{u^- - \Delta}^{u^-} \Delta\mathrm{d}P$$

$$\leq \Delta\epsilon.$$

Hence

$$\Delta(\tau - F(u^-)) \leq \mathbb{E}_U[l_\tau(u - \Delta, U)] - \mathbb{E}_U[l_\tau(u, U)] \leq \Delta(\tau - F(u^-) + \epsilon).$$

Then

$$\tau - F(u^-) \leq \liminf_{\Delta \to 0^+} \frac{\mathbb{E}_U[l_\tau(u - \Delta, U)] - \mathbb{E}_U[l_\tau(u, U)]}{\Delta}$$

$$\leq \limsup_{\Delta \to 0^+} \frac{\mathbb{E}_U[l_\tau(u - \Delta, U)] - \mathbb{E}_U[l_\tau(u, U)]}{\Delta}$$

$$\leq \tau - F(u^-) + \epsilon.$$

Since $\epsilon$ can be arbitrarily small, the limit exists, and

$$\partial_- \mathbb{E}_{U \sim F}[l_\tau(u, U)] = -\lim_{\Delta \to 0^+} \frac{\mathbb{E}_U[l_\tau(u - \Delta, U)] - \mathbb{E}_U[l_\tau(u, U)]}{\Delta} = F(u^-) - \tau.$$

The other side of semi-derivative can be derived in a similar way, which we omit for simplicity. □

**Lemma 15.** *For any $a_1, a_2, b_1, b_2 \in \mathbb{R}$, we have*

$$|\max\{a_1, b_1\} - \max\{a_2, b_2\}| \leq \max\{|a_1 - a_2|, |b_1 - b_2|\}.$$

*Proof of Lemma 15.* If $a_i \geq b_i$ (or $a_i \leq b_i$) simultaneously for $i = 1, 2$, the claim is straightforward. Without loss of generality, we assume $a_1 < b_1 < b_2 < a_2$. Then

$$\text{LHS} = a_2 - b_1 \leq a_2 - a_1 = \text{RHS}.$$

□

*Proof of Lemma 13.* As for part (a), we notice that

$$\mathbb{E}\left[\frac{1}{1 + \zeta}\right] = \sum_{j=0}^n \frac{1}{j+1} C_n^j p^j (1-p)^{n-j}$$

$$= \frac{1}{(n+1)p} \sum_{j=0}^n C_{n+1}^{j+1} p^{j+1} (1-p)^{n-j}$$

$$\leq \frac{1}{(n+1)p} \sum_{j=0}^{n+1} C_{n+1}^j p^j (1-p)^{n-j+1}$$

$$= \frac{1}{(n+1)p} (p + 1 - p)^{n+1} = \frac{1}{(n+1)p},$$

where $C_n^j$ is the number of combinations of $(n, l)$.

Since $\frac{1}{k} \leq \frac{2}{k+1}$ for any $k \geq 1$, we have

$$\mathbb{E}\left[\frac{1}{\zeta} \mathbb{1}\{\zeta > 0\}\right] \leq \mathbb{E}\left[\frac{2}{1 + \zeta}\right] \leq \frac{2}{(n+1)p}.$$

For part (b), we prove its symmetric argument that $\forall r > np$,

$$\mathbb{P}(\zeta \geq r) \leq \frac{rq}{(r - np)^2},$$

where $q = 1 - p$.

The probability ratio of two adjacent value is

$$\frac{\mathbb{P}(\zeta = r + 1)}{\mathbb{P}(\zeta = r)} = \frac{(n - r)p}{(r + 1)q}$$

$$\leq \frac{(n - r)p}{rq}$$

$$= 1 - \frac{r - np}{rq}$$

$$< 1.$$

Hence for any $k \geq r$, we have

$$\mathbb{P}(\zeta = k) \leq \mathbb{P}(\zeta = r) \cdot \left(1 - \frac{r - np}{rq}\right)^{k-r}.$$

Summing all these $k$'s from $r$ to $\infty$, we have

$$\mathbb{P}(\zeta \geq r) = \sum_{k=r}^{\infty} \mathbb{P}(\zeta = k)$$

$$\leq \mathbb{P}(\zeta = r) \sum_{k=r}^{\infty} \left(1 - \frac{r - np}{rq}\right)^{k-r}$$

$$= \mathbb{P}(\zeta = r) \frac{1}{1 - \left(1 - \frac{r-np}{rq}\right)}$$

$$= \mathbb{P}(\zeta = r) \frac{rq}{r - np}.$$

From a similar argument of inspecting the probability ratio between $\lceil np \rceil \leq k \leq r$, we know that

$$\mathbb{P}(\zeta = k) \geq \mathbb{P}(\zeta = r).$$

There are at least $r - \lceil np \rceil + 1 \geq r - np$ such $k$'s (including $r$ itself), so we have

$$1 \geq (r - np)\mathbb{P}(\zeta = r),$$

which leads to the conclusion that

$$\mathbb{P}(\zeta \geq r) \leq \frac{rq}{(r - np)^2}, \quad \forall r > np.$$

If we have $r < np$, then we repeat the above arguments by replacing $r$ with $n - r$ and reversing $p$ and $q$, we can prove

$$\mathbb{P}(\zeta \leq r) \leq \frac{(n - r)p}{(np - r)^2}, \quad \forall r < np.$$

$\square$

