# OpenReview forum: "Distribution-Free Model-Agnostic Regression Calibration via Nonparametric Methods"
_NeurIPS.cc/2023/Conference — NeurIPS 2023 poster_

### Official Review · Reviewer_bxwF · 2023-07-05

**Soundness:** 3 good
**Presentation:** 3 good
**Contribution:** 1 poor
**Rating:** 4
**Confidence:** 5

**Summary:**

This paper addressed the uncertainty quantification problem in regression models, specifically focusing on individual calibration to characterize prediction model quantiles. To overcome these limitations, they proposed simple nonparametric calibration methods that are both computationally efficient and statistically consistent. Their approach provides insights into individual calibration possibilities and establishes upper and lower bounds for calibration error. They attempted to advance existing theoretical analyses by combining nonparametric and parametric techniques, offering new perspectives on regression calibration regarding the curse of dimensionality and reconciling previous findings on individual calibration impossibility.

**Strengths:**

The paper is overall well written and the guarantees given are technically sound. The authors have demonstrated the effectiveness of their methods through several experiments. The ideas developed are novel to the best of my knowledge but their practicality is questionable. There are many advancements in the field of conformal inference that give similar guarantees with minimal assumptions. I think the paper would benefit from comparing their developed method in varied settings with the existing approaches in conformal inference to prove its efficacy.

**Weaknesses:**

In this paper, the authors tackle the uncertainty quantification problem for regression models, focusing on individual calibration. While the proposed nonparametric calibration methods and the accompanying analysis present some interesting ideas, I have serious concerns about the authors' familiarity with previous works in the field, as well as the lack of necessary citations to support their claims.

One significant issue is the absence of references to previous research on conditional conformal prediction, which is a relevant and well-established framework in uncertainty quantification. The authors should have acknowledged and discussed how their proposed methods relate to or differ from existing conditional conformal prediction approaches. Failure to do so undermines the paper's novelty and raises doubts about the authors' understanding of the current state-of-the-art in the field. There have been various works in this area namely:
1. "Conformalized Quantile Regression" by "Yaniv Romano, Evan Patterson, Emmanuel J. Candès",
2. "Improving conditional coverage via orthogonal quantile regression" by "S Feldman, S Bates, Y Romano"
3. "Class-Conditional Conformal Prediction With Many Classes " by "T Ding, AN Angelopoulos, S Bates, MI Jordan, RJ Tibshirani"
4. "Conformal prediction with conditional guarantees" by "I. Gibbs, J. J. Cherian, E. J. Candès "
5. "Knowing what you know: valid and validated confidence sets in multiclass and multilabel prediction" by "M Cauchois, S Gupta, JC Duchi"

The proposed method also suffers from the curse of dimensionality and the authors have only included experiments with several UCI datasets which are relatively low-dimensional. Whereas on the other hand, advancement in the field of conformal inference has produced methods that seamlessly adapt to very high dimensional datasets like ImageNet, MNIST, etc.

Further, the theoretical guarantees in the paper rely on very strong assumptions, i.e., Assumption 1 of the paper, unlike previous works in this field.




**Questions:**

I do not have further questions, all the concerns are summarised in the "weakness" and "limitation" sections.

**Limitations:**

 The proposed method suffers from the curse of dimensionality as I mentioned before and the method relies on very strong assumptions. The authors have not implemented their proposed method on high dimensional datasets.

---

> ### Author Rebuttal · Authors · 2023-08-09
>
> We thank the reviewer for the sincere comments and for pointing out our missing literature. In the past week, we spent quite much time working on these papers and related literature. We’d like to summarize our findings below. We hope this better clarifies our positioning, and we look forward to having more discussions with you and other reviewers in the coming week.
>
> Positioning of our work:
>
> The reviewer raised a question on whether such algorithms (with the goal of individual calibration) are practical for high-dimensional setting. As noted in our paper, our work is more aimed at providing a positive result for individual calibration, given that the existing results on individual calibration for regression problems are largely negative. Let’s separate the regime into low-dimensional and high-dimensional.
>
> - Low-dimensional regime: after reading our work, we guess probably all the reviewers agree that individual calibration is to some extent achievable under the lower-dimensional regime; yet this is a fact that is not acknowledged by the existing literature. Importantly, we also justify the necessity of striving for such an individual calibration by the pinball loss from the downstream task such as inventory management or newsvendor problem (Proposition 2), which serves as a motivating example for many calibration papers. These two combined imply that being intimidated by the hardness of individual calibration is unnecessary and cost-ineffective.
>
> - High-dimensional regime: Our paper answers the question of “when individual calibration is achievable”, but we didn't say that individual calibration is always achievable. For high-dimensional data, we agree with the reviewer that individual calibration can be a goal too ambitious to target. Yet, on the positive end, as in Theorem 3 of our paper (the proof of which is technically very straightforward), if the high-dimensional covariates have low-dimensional statistics that induce some conditional independence, then individual calibration can be achieved with the low-dimensional statistics. Practically, the low-dimensional statistics can be obtained by training a neural network with a small number of neurons in the second-last layer. There has been a line of literature studying the ability of feature extraction for neural network models; Theorem 3 says that such development can be useful in achieving individual calibration in a high-dimensional regime.
>
> - The reviewer mentioned high-dimensional datasets such as MNIST and ImageNet. Both datasets are tested only as a classification task by the existing literature on conformal prediction/calibration, while our paper studies on regression problems. The eight datasets in the numerical experiments are chosen because they are the standard ones used in most (if not all) regression calibration papers. For demonstration purposes, we include a new dataset of Yacht which is of higher dimension (d=100, n=2000), and observe similar advantages of our algorithm compared to the benchmarks. The 100-dimensional vector might not be as large as the MNIST or ImageNet, but when working on an even higher dimension, the second-last layer of the neural network should have a dimensionality of a similar magnitude.
>
> Literature on conformal prediction:
>
> We appreciate the reviewer for bringing the papers to our attention. We believe, and as noted by other reviewers, our paper includes a much longer literature list than a normal conference paper; we didn’t miss these papers deliberately (Also, 2 papers mentioned by the reviewer are in fact posted after NeurIPS 2023 submission deadline). Also, as noted in our paper, we found researchers from different communities, conformal prediction, calibration, and quantile regression, working on similar or even identical problem setup but don’t acknowledge much each other’s work. For example, the existing calibration literature doesn’t cite much the papers on conformal prediction, and vice versa. To this end, we hope our work can make at least some minimal effort in syncing the understanding and awareness across communities, and in our paper, we did mention the perspective of conformal prediction throughout the presentation of our results. Hence, we read these conformal prediction papers with great interests and will include a discussion on this line of works in the next version of our paper. We defer a detailed discussion to the Author Rebuttal.
>
> We also want to bring up one thing to the reviewer. The majority of work in conformal prediction follows a two-step procedure, also known as a split procedure, that first fits a mean prediction model and then calibrates the prediction error. Meanwhile, the majority of work in calibration literature follows a one-step procedure, that usually appends a calibration penalty to the mean prediction loss in the loss function. The two-step split procedure has the apparent advantage of avoiding overfitting and thus avoids too small confidence interval as the one-step procedure. Our paper provides a second justification for this two-step procedure (See our response to aPS1), which we believe complements the existing works on conformal prediction.
>
> Lipschitzness assumption:
>
> The Lipschitzness condition in our paper helps answer one key question of conformal prediction. It should be regarded as the minimum assumption to avoid violating requirements (i) and (iii) (see the Author Rebuttal for the requirements), where the term “minimum” is proved by our Theorem 2 and Theorem 6. Theorem 6 states that without such a Lipschitz condition, any estimator, even if asymptotically consistent, can suffer from an arbitrarily low convergence rate. In other words, it is an unavoidable cost to achieve a meaningful (iii). Theorem 2, on the other hand, proves that the minimax lower bound under a Lipschitz condition is matched (up to some poly-logarithmic factors) by our algorithm.

---

> > ### Comment · Reviewer_bxwF · 2023-08-14
> >
> > We thank the authors for the clarifications.
> >
> > "Also, as noted in our paper, we found researchers from different communities, conformal prediction, calibration, and quantile regression, working on similar or even identical problem setup but don’t acknowledge much each other’s work."-- I am sorry but I am not convinced by these line of reasonings. I believe we are posing a problem relevant to the scientific community and proposing a solution that works and somewhat better than existing methods (thus, adding to the novelty). The field of conformal inference has flourished in the recent times and has shown exceptional promise in solving the posed problem both in regression and classification settings. As for split conformal, we have methods like Jackknife + (Barber et al. 2022) which seamlessly avoids data splitting. Given the generalizability of conformal inference type methods across different dimensionality of datasets, types of problems (regression and classification), and computational efficiency, the paper stands incomplete without proper comparison.

---

> > > ### Author Response · Authors · 2023-08-15
> > >
> > > We thank the reviewer for the follow-up. In writing the paper and the previous responses/discussions, we try to follow two principles: (i) open-mindedness; (ii) technically rigorous arguments. We understand that this can be a busy week for all the reviewers, but we appreciate it very much if the reviewer can spend some time reading our paper and the previous responses.
> > >
> > > Our apologies for writing a lengthy response as before. For the sake of saving time, we provide a TL;DR version for the literature mentioned by the reviewer below:
> > >
> > > - Paper 3 and Paper 4 are posted on ArXiv after the NeurIPS 2023 submission deadline. We are not sure if these two papers are also under review at NeurIPS 2023.
> > >
> > > - Even so, we aim for open-mindedness and will include these two papers in future versions of our paper. In our previous response, we discussed that Paper 3 and Paper 4 are not aiming for an individual coverage guarantee, which differs from our positioning of individual calibration.
> > >
> > > - Paper 1 and Paper 5, together with the new paper Jackknife + (Barber et al. 2022), mentioned by the reviewer, all aim for a marginal calibration guarantee. For the newly mentioned paper (Barber et al. 2022) apart from the marginal calibration guarantee v.s. our individual calibration guarantee, the proposed method can be computationally costly compared to our method because it is a Jackknife-based approach.
> > >
> > > - Paper 2 aims for individual calibration but requires much stronger conditions for their theoretical results (See the last few paragraphs in our Author rebuttal).
> > >
> > > We see all above as “facts” that are supported by technically rigorous arguments. We are happy to follow up with additional technical discussions if there are any confusions/comments with these facts.
> > >
> > > We’d like to separate “facts” from “opinions”, such as the following one which is disagreed by the reviewer:
> > >
> > > "Also, as noted in our paper, we found researchers from different communities, conformal prediction, calibration, and quantile regression, working on similar or even identical problem setup but don’t acknowledge much each other’s work."
> > >
> > > This is our opinion, which of course, can be agreed or disagreed with by other researchers. We have such an impression/opinion because of the fact that for the 50+ papers in our reference list, the calibration papers mostly don’t mention the work on conformal prediction, while the conformal prediction papers don’t cite the calibration papers. However, even before this discussion, we try to maintain an open-mindedness and draw connections between our results and the conformal prediction literature in our paper.
> > >
> > > We thank all the reviewers and ACs for taking the time to read our responses.

---

> > > > ### Comment · Reviewer_bxwF · 2023-08-17
> > > >
> > > > I appreciate the authors for their follow up. However I maintain my rating for the following reasons:
> > > >
> > > > 1) "Paper 3 and Paper 4 are posted on ArXiv after the NeurIPS 2023 submission deadline." Yes and in that case you do not need to cite these works. My intention to suggest a list of few papers ignoring many others was to draw attention of the authors that active research on similar problem is undergoing and some of them should be added as a baseline.
> > > >
> > > > 2)  Jackknife + (Barber et al. 2022): This paper has faster implementation (see their K-fold cross validation version) which drastically improves computational efficiency.
> > > >
> > > > 3) Indeed finite sample conditional coverage is impossible without further assumptions but several attempts have been made in the field of conformal inference. As mentioned in Author's rebuttal above, paper 2 comes close to their paper and their is no discussion or benchmarking in experiments with the submitted paper.
> > > >
> > > > 4) An essential focal point within this domain pertains to the size of sets. Extensive set sizes tend to lack informative value, constituting a significant metric frequently employed for appraising the effectiveness of prevalent predictive inference techniques highlighted across numerous conformal inference publications. Regrettably, this metric is absent in the current paper. While I acknowledge potential constraints such as the absence of cross-community citations, addressing these omissions and conducting thorough benchmarking would undoubtedly elevate the scholarly contribution.
> > > >
> > > > 5) Paper 2 works under a different set of assumptions but is easily adaptable to a wide range of real datasets and particularly high dimensional set up as established in their experimental set up. While the proposed method in the submitted paper again relies on an underlying low dimensional structure for a high dimensional feature set. I am finding it hard to believe the scope, applicability of the proposed method, and efficacy in terms of set sizes in wide range of real datasets.
> > > >
> > > > I again appreciate the authors for their work and I think the paper would strongly benefit with considerations of relevant work from the field of conformal inference and validate the novelty of contributions particularly in real data because these methods are widely used in many field of applications and have shown promising results.

---

> > > > > ### Author Response · Authors · 2023-08-21
> > > > >
> > > > > We thank the reviewer for the helpful comments. We agree with the reviewer and appreciate the conformal prediction community for their efforts and results, and we find the papers mentioned very inspiring. In the past few days, we implemented numerical experiments to compare our algorithm with the algorithm in Paper 2 on the 6 datasets that are of the highest dimensions among the datasets in Paper 2. Such conformal prediction benchmarks and detailed experiments will be included in our further versions.
> > > > >
> > > > > 1. We apologize for missing some of the conformal prediction literature, but indeed we tried our best for the literature review. We are willing to add more discussions as mentioned in our previous responses in the future. Among the conformal prediction literature, we aim to address the minimum assumption to keep both the individual calibration (or coverage) and the finite sample guarantee, reconciling the impossibility results for quantile calibration (see our Appendix B) as well as the impossible triangle in the field of conformal prediction. Theoretically, we don’t find any paper in the literature on conformal prediction that gives a better result with comparable or weaker assumptions (including the papers mentioned by the reviewer). We refer to the previous Author Rebuttal for details.
> > > > >
> > > > > 2. As for the (practical) performance metrics, although the quantile calibration task is very closely related to the conformal prediction task, the goals are still different. Conformal prediction aims to provide as sharp as possible prediction sets that cover the true outcomes with desired rates. While a precise quantile prediction guarantees the coverage rate, such a covering band may not be the sharpest. As is discussed in Appendix A, the additional sharpness regularization term could harm the goal of precise quantile prediction. For example, people may select 0.05 and 0.95 quantiles to construct a 0.9 coverage band, but the sharpest covering band probably would deviate from such quantiles. But a high-quality quantile prediction alone is of independent interest for many downstream tasks, such as the newsvendor problem. This points to a difference between the objectives of calibration and conformal prediction. As a result, the popular performance metrics applied in conformal prediction such as the average interval length and the coverage rate are not direct measurements for the quantile calibration problem. Some other measurements, for example, the independence between the coverage indicator and the band length in Paper 2, are only a necessary condition of the quantile calibration. We henceforth consider only the explicit measurements rather than such implicit ones in our paper.
> > > > >
> > > > > 3. Numerically, to further illustrate the issue of high-dimensionality, we selected the 6 highest-dimensional datasets that appear in Paper 2 and applied our algorithm and Paper 2’s Pearson-correlation-regularized algorithm. First, we note that the first step regression of our algorithm doesn’t have to be the mean regression but can be any regression algorithm. We apply a quantile regression algorithm as the first step, and then recalibrate the initial result using our nonparametric estimator. As for the metric, we select (a 90% confidence interval variant of) the Adversarial Group Calibration Error (AGCE, defined in Appendix D.1) to show calibration error in the worst calibrated part of the data. We list the results as follows (Paper 2’s OQR the former, and the latter ours):
> > > > > meps_19:
> > > > > 0.071,
> > > > > 0.03
> > > > > meps_20:
> > > > > 0.059,
> > > > > 0.031
> > > > > meps_21:
> > > > > 0.043,
> > > > > 0.036
> > > > > facebook_1:
> > > > > 0.041,
> > > > > 0.022
> > > > > facebook_2:
> > > > > 0.011,
> > > > > 0.018
> > > > > blog_data:
> > > > > 0.03,
> > > > > 0.024.
> > > > > The numerical performance shows the advantage of our algorithm. Of course, AGCE is a conditional/individual performance measure; for marginal calibration objectives or other datasets, conformal prediction methods may have an advantage. Just like the general ML problem, we don’t expect a single ML model that performs universally well, but we do believe our algorithm provides a simple and efficient complement to the existing methods.
> > > > >
> > > > > 4. As for the high-dimensional cases, we here make some further explanations. On one hand, the belief that high-dim data can be represented by lower-dim features is common in modern machine learning. Both computer scientists and statisticians are making efforts to extract low-dim features from the original high-dim datasets. On the other hand, the tremendous impact of reducing the Lipschitz coefficient $L$ partly explains why our algorithm works even in the presence of high dimensionality (and also why there is a splitting procedure in the split conformal prediction). As mentioned by the reviewer, there are conformal prediction algorithms that don’t follow the split protocol, but to our knowledge, we provide a first explanation/justification for those split conformal prediction algorithms.
> > > > >
> > > > > We thank the reviewer again for their helpful comments, which we believe consolidate the overall positioning of our paper.

---

### Official Review · Reviewer_2q3v · 2023-07-07

**Soundness:** 3 good
**Presentation:** 3 good
**Contribution:** 3 good
**Rating:** 6
**Confidence:** 3

**Summary:**

This paper studies uncertainty quantification for the regression problem. In particular,
it considers the estimation of conditional quantiles (of the residuals) via the kernel method. The convergence
rate of the proposed estimator is established, along with a matching lower bound. The proposed
method is evaluated on multiple datasets and compared with other candidate methods.

**Strengths:**

The paper considers an interesting problem; the examples
for showing the unexpected results of existing methods are motivating;
the
solution provided has solid theoretical properties
and show satisfactory empirical performance in
numerical experiments.

**Weaknesses:**

I was wondering about the position of this paper in the line of works
of conditional quantile regression (e.g., Takeuchi et al. (2006); Steinwart et al. (2011)).
A discussion in this direction will be appreciated.

References:

Takeuchi, Ichiro, et al. "Nonparametric quantile estimation." (2006).

Steinwart, Ingo, and Andreas Christmann. "Estimating conditional quantiles with the help of the pinball loss." (2011): 211-225.

**Questions:**

1. As mentioned above, I wonder how this work
compares with the line of works of (conditional) quantile regression.
2. It might be helpful to also show in the simulations
the results if one directly estimates the conditional quantiles
of $Y$.
3. I wonder if the proposed method can be used within the
framework of conformal inference and achieve distribution-free
marginal calibration as well.

**Limitations:**

 The authors have adequately addressed the limitations.

---

> ### Author Rebuttal · Authors · 2023-08-09
>
> We thank the reviewer for appreciating our work and raising inspiring questions.
>
> Direct estimation:
>
> We include more experiments using that directly estimate the conditional quantiles by optimizing pinball loss as the attachment. It does show the advantage of the two-step procedure. We also provide more theoretical explanations for the advantage of this decomposition or two-step procedure in our response to Reviewer aPS1, and we are grateful if you have time and are interested in reading the explanations therein.
>
> Marginal calibration for conformal prediction:
>
> In this work, we focus mainly on the individual calibration objective; and our main motivation when initiating the project is in fact to provide a positive result for the pessimism of individual calibration mentioned by the existing works. For marginal calibration, there are many existing methods in the literature of conformal prediction that enjoy both successful empirical performance and a nice theoretical guarantee (see also our response to Reviewer bxwF). For our method, if one aims for a marginal calibration guarantee, then a simple way to modify Algorithm 1 would be to remove the kernel weighting and assign a uniform weight to all the samples. The marginal calibration guarantee can then be derived under fewer assumptions (without Assumptions 1 (a) and (c)).
>
> Positioning against (conditional) quantile regression:
>
> First, we’d like to thank the reviewer for bringing up these two papers. We believe we have done an exhaustive search over the literature view part of the papers cited in our paper, but we still miss this line of works on conditional quantile regression. As we noted in our paper and our responses to other reviewers, it seems to us that several communities are working on the same problem but aren’t aware of/don’t cite each other. In this light, we have read the papers with great interest and will definitely include them in the next version of our paper.
>
> Both papers utilize the kernelized method and directly predict the quantile function (one-step). Essentially, methods from both papers search for a quantile prediction function over the RKHS function space $\mathcal{H}$. Apart from the comparison between the one-step method and our two-step method in our response to review aPS1, we discuss the positioning of our results against these two works in two additional aspects:
>
> - Theoretically, Takeuchi et al. (2006) derived a performance guarantee for the conditional quantile estimator. Their derivation is based on a Rademacher complexity bound over the function class with a bounded $\mathcal{H}$-norm, and to ensure a consistency result, it requires the true conditional quantile function to have a bounded $\mathcal{H}$-norm. This is in parallel with our Lipschitz class (Assumption 1 (a)). The Lipschitz function class and the bounded $\mathcal{H}$-norm function class overlap but don’t contain each other. Steinwart and Christmann (2011) analyze the same algorithm but adopt a different approach. Instead of imposing a bounded $\mathcal{H}$-norm, they impose a decay rate for the eigenvalues of kernel integral operator, alongside some other assumptions on the underlying distribution (on a minor note, their paper missed a condition in their theorem statement that the chosen kernel should lead to an RKHS dense in $L_1$ space). In short, both papers, together with ours, strive to fight against the impossibility of individual calibration by imposing conditions on the true quantile function so that it belongs to a certain function class. These two works focus on RKHS space while we focus on Lipschitz space; generally, we find the conditions imposed not quite comparable to each other. For the Lipschitz function class, to the best of our knowledge, our work is the first result to provide an individual guarantee.
>
> - Empirically/computationally, our algorithm features an analytical solution and thus is very simple and efficient to implement, while both papers rely on solving a kernelized learning problem that doesn’t scale well with sample size or dimension. For Steinwart and Christmann (2011), it is more of a theoretical work and doesn’t provide any numerical experiments. From a practical viewpoint, our method/framework is more about (i) justifying the widely-adopted two-step calibration procedure and (ii) providing a simple solution and positive result for individual calibration. Though our work and these two papers aim for the same individual guarantee, we don’t quite see our framework as a competitor to them. Because at the end of the day, one can adopt the kernelized method for the mean prediction part (in replacement of a linear model or NN) and/or the error quantile prediction part (in replacement of the simple nonparametric estimator) of our framework.
>
> We hope our response addresses the raised questions. If there are any follow-up questions/concerns, we will get back to you timely in the following discussion week.

---

> > ### Comment · Reviewer_2q3v · 2023-08-20
> > **Response to the authors**
> >
> > I thank the authors for the response and the comparison, and I look forward to seeing these contents added to the paper.

---

### Official Review · Reviewer_oVNP · 2023-07-08

**Soundness:** 3 good
**Presentation:** 3 good
**Contribution:** 3 good
**Rating:** 6
**Confidence:** 3

**Summary:**

The paper considers the uncertainty quantification problem for regression models. First, they proposed an algorithm for simple nonparametric quantile estimator. Then, they further proposed the nonparametric regression calibration algorithm. They also provide theoretical analysis of the proposed algorithms and implications.

**Strengths:**

1)	The paper is well-organized and well-written.
2)	The paper provides several theoretical results and implications of the theory.
3)	The paper includes extensive experiments.

**Weaknesses:**

1)	For Algorithm 1, do other kernels work for the proposed algorithm? Are there any requirements for the kernels?
2)	For Algorithm 2, could the split proportion be different than half and half? The proportion would influence the results. Are there any experimental results to check the effect of the proportion?


**Questions:**

1)	For Algorithm 1, do other kernels work for the proposed algorithm? Are there any requirements for the kernels?
2)	For Algorithm 2, could the split proportion be different than half and half? The proportion would influence the results. Are there any experimental results to check the effect of the proportion?

---

> ### Author Rebuttal · Authors · 2023-08-09
>
> We thank the reviewer for the questions and comments.
>
> The choice of the kernel function:
>
> This simple nonparametric method performs rather robustly with respect to the choice of the kernel function. Essentially, all standard choices of kernels specify a localized weighting regime for error quantile estimation and enjoy similar or even identical theoretical guarantees. Numerically, we presented the naive kernel and the Gaussian kernel (RBF) in the paper but we also tried out the Laplace RBF kernel, normalized triangle kernel, and higher-order kernels on different tasks. And they all give similar performances. We didn’t stress much on the point in the submitted version, but we will include more information about this aspect in the next version of our paper.
>
> Data splitting scheme:
>
> We remark that the split between the mean estimation data and the error quantile estimation data is quite flexible and it doesn’t have to be 50-50. We are sorry for causing the impression of a half-and-half split. The split ratio in fact trades off between learning a good mean prediction model and achieving a decent estimation of quantiles. In the numerical experiments, we take a grid search for ratios between 2:8 to 8:2, and the best proportion is picked based on the performance of the validation set. For some datasets, the best proportion happens to be half-and-half, but this is not always the case. Interestingly, many works on split conformal prediction do recommend such a 50-50 ratio but didn’t provide much theoretical justification. We will update our paper with the point and look forward to see more future investigations on this aspect.
>
> We hope our response addresses the raised questions. If there are any follow-up questions/concerns, we will get back to you timely in the following discussion week.

---

### Official Review · Reviewer_aPS1 · 2023-08-01

**Soundness:** 3 good
**Presentation:** 4 excellent
**Contribution:** 3 good
**Rating:** 8
**Confidence:** 2

**Summary:**

This paper proposes a new method for quantile regression and for calibrating prediction intervals in regression. The paper first proposes a simple quantile regression method and shows that this method estimates the true quantile curve at the minimax-optimal rate. For calibrating prediction intervals, the basic idea is to 1) decompose the conditional distribution of the response variable into a conditional mean + noise term, 2) estimate the conditional mean, and 3) apply the above quantile regression method to the distribution of residuals (i.e., respose minus estimated conditional mean). The approach is agnostic to the method used to fit the conditional mean itself, and the experiments demonstrate that the proposed method (as well as a supplementary method that also applies dimension reduction) performs well using feed-forward and recurrent neural nets, as well as random forests.

**Strengths:**

The paper is quite easy to follow. The proposed method seems both simple and effective, and makes very weak assumptions. Figure 1 makes the advantage of individual calibration very clear, and the theoretical guarantees are hence both useful and impressive in light of existing results on the impossibility of individual calibration. The experiments are also fairly thorough and well-presented.

**Weaknesses:**

1) Table 1: The "Ours Best?" column seems misleading, because it is taking the best of two different methods (NRC and NRC-DR). In several cases, only one of the proposed methods performs best, while the other method performs worse than competitors. However, in practice, one must typically pick one method before knowing which of the two will perform better. So, this column is not really informative of how the proposed methods would perform in practice. I suggest either removing this column. Perhaps a vertical rule could be added to distinguish the current paper's methods from previous methods.

2) The introduction is quite long, and it is not clear to me whether all of this information should be presented so early in the paper.
For example, the discussion on Page 2 about "individual calibration" is unclear since "individual calibration" isn't defined until Page 3. Much of the content could also be refactored into a specific "Related Work" section.

**Questions:**

1) One of the paper's key ideas is to decompose the prediction interval problem into a mean regression problem and a noise quantile prediction problem. Are there any cases where this decomposition would fail, or would be expected to perform worse than a method that learns the regression quantiles directly?

2) Relatedly, how does the performance of the mean estimator affect the performance of the prediction interval (e.g., if the mean estimator under- or overfits)?

**Limitations:**

The paper is fairly clear about its limitations, such as the gap between the theoretical performance (which suffers from the curse of dimensionality) and the strong empirical results.

---

> ### Author Rebuttal · Authors · 2023-08-09
>
> We thank the reviewer for the feedback and the raised questions. We believe clarifying these questions improves the positioning of our work.
>
> Decomposition/two-step procedure:
>
> As noted by the reviewer, our paper adopted a two-step approach which first predicts the mean and then calibrates the quantile of the noise. Empirically, we do find that the two-step procedure is better than the one-step procedure that directly predicts the quantile of $Y|X$ (see the attachment for the new experiment). Theoretically, this has a nice justification too. Specifically, from the minimax lower bound in Theorem 2, the problem’s hardness depends on the Lipschitz constant $L$, the sample size $n$, and the feature dimension $d$. The two-step procedure subtracts an initial regression model $\hat{f}(X)$ from the label $Y$: in the following, we argue why such subtraction can effectively reduce the Lipshitz constant.
>
> Generally speaking, the conditional expectation function $E[Y|X]$ is highly related and very likely to wax and wane together with the quantile function $Q_{\tau}[Y|X]$. In this light, subtracting the conditional expectation function $E[Y|X]$ can very likely smooth out the quantile function $Q_{\tau}[Y|X]$, resulting in a smaller Lipschitz constant. In metaphor, this is quite like the method of controlled variates in Monte Carlo simulation which introduces a controlled random variable that is correlated with the target random variable so as to reduce the variance of the estimator. Here, the conditional expectation function $E[Y|X]$ works as the “controlled variate” for the original “target variate” of $Q_{\tau}[Y|X]$ (the Lipschitz constant $L$ here corresponding to the variance to be reduced in Monte Carlo simulation). Of course, there might exist data distributions where $E[Y|X]$ and $Q_{\tau}[Y|X]$ are completely unrelated, and in that case, the two-step procedure doesn’t guarantee an improvement over the one-step procedure. Also, in reality, we will only use the fitted conditional expectation, $\hat{E}[Y|X]$ (from the mean prediction model). Regardless of overfitting or underfitting, as long as the function $\hat{E}[Y|X]$ behaves relatedly with the quantile function $Q_{\tau}[Y|X]$, it can effectively reduce the Lipschitz constant in logic. In the extreme case, the Lipschitz constant L=0 for the error quantile function $Q_{\tau}[Y-E[Y|X]|X]$ corresponds to the case of homoscedasticity, which is still reasonable for some data distribution; but meanwhile, we can’t really expect the original $Q_{\tau}[Y|X]$ to have $L=0$, i.e., to be a constant function.
>
> Paper writing and introduction:
>
> We thank the reviewer for the kind advice. We agree with the comments on the column of “Ours best” and will adjust it accordingly in the next version of our paper. Also, for the introduction, we apologize for the usage of concepts before formal definitions and will revise it.
>
> In terms of the length of our introduction, it is partly due to the nature of the studied problem that it lies on the intersection of calibration, quantile regression, and conformal prediction, spreading over several fields of machine learning and statistics. The ML literature has a related but different taste than the statistics literature. The calibration literature, which originates from the community of machine learning, is often based on modern machine learning models such as deep neural networks. It usually proposes models at the price of sacrificing some theoretical rigorousness. Concurrently, the statistics literature develops concepts that always have theoretical guarantees. But sometimes it is not so straightforward: some recent advances in conformal prediction claim their excellence in dealing with heteroscedasticity, while still lacking a finite-sample theoretical guarantee on reaching such an individual coverage. For more discussion on this, we refer to our response to Reviewer bxwF. We skip some quantile regression and conformal prediction literature reviews that are less relevant but still result in a long introduction. Moreover, papers from different communities usually don’t cite each other much, though they are working on a similar or even the same problem. In this light, we hope our work can make at least some contribution to improving the awareness and synchronizing the language/understanding for researchers from different communities on this same problem.
>
> We hope our response addresses the raised questions. If there are any follow-up questions/concerns, we will get back to you timely in the following discussion week.

---

> > ### Comment · Reviewer_aPS1 · 2023-08-16
> >
> > Thanks to the authors for their response.
> >
> > Regarding the decomposition: I understand why this approach reduces the Lipschitz constant of the estimand and thereby reduces the difficulty of estimation. My question was whether there are any counterexamples where this would be expected to perform poorly (or less well than a one-step procedure). If so, the paper would be made stronger and clearer by discussing such a counterexample.
> >
> > Regarding the writing of the introduction, I understand the reason for the length (and I appreciated what felt to me like a thorough literature review). My suggestion was that it could be better organized for the reader (e.g., by using some more subsection/paragraph headers, or by moving some of the content that isn't needed to understand the proposed method into a "Related Work" section). Although it's important to point out gaps in the existing literature (e.g., in a "Related Work" section), I feel that the basic motivation for proposing an approach shouldn't depend so heavily on the existing literature (which is always changing).
> >
> > I don't necessarily expect the authors to reply to the above points in the discussion period -- just some things to think about as they continue to revise the paper. Overall, I still feel this paper gives a simple, effective, and novel approach to an important problem, backed by solid experiments and theory, and so I intend to keep my score of 8.
> >
> > That said, I am not up-to-date on the conformal inference literature (the main reason for my low confidence of 2), and I defer to Reviewer bxwF on whether critical papers are missing in this regard (beyond what the authors could easily add to the camera-ready version).

---

> > > ### Author Response · Authors · 2023-08-21
> > >
> > > We thank the reviewer for raising the points; we will continue thinking about them and include more content in these aspects in the future version of our paper.

---

### Author Rebuttal · Authors · 2023-08-09

We thank the reviewers for taking time to read our papers and for all the helpful feedback. We look forward to more follow-up discussions in the coming week.

We provide an individual response to each reviewer, and we'd like to use the extra space below to discuss the assumptions of our paper in response to the questions raised by Reviewer bxwF on assumptions and comparisons against conformal prediction literature. Sorry for the caused inconvenience and confusion.

The only assumption made in our paper is Assumption 1. Part (a) and part (b) are standard and critical in achieving individual calibration guarantees. Part (c) is used to obtain the finite-sample convergence rate. Essentially, a by-product of our analysis derives a concentration argument for quantile estimators. Part (c) is indeed milder than the existing literature on quantile bandits and nonparametric density estimation that also derives such quantile concentration results. We believe this part of the assumption compared to the literature is of independent interests and a contribution itself. The key is that we adopt a new analysis approach that combines both ideas for parametric and nonparametric analysis. We refer to more discussions in Line 210-219 of our paper.

Now we proceed with a detailed discussion of the 5 papers mentioned by the reviewer. After careful examination, we don’t believe any of this work (or other works mentioned in our paper’s reference list and our response to the reviewers) provides a comparable result as ours under a comparable or weaker assumption.

All 5 listed papers are developed in the area of conformal prediction, where the goal is to give a coverage set that contains the true label with a certain probability. Thus the goal of conformal prediction is no harder than our quantile prediction task since correct quantiles induce desired coverage sets. Among these 5 papers, Paper 3 and Paper 5 are for classification problems, and the remaining 3 papers are for regression problems as ours.

First, we rephrase the “impossible triangle” summarized in Paper 4: (i) a conditional/individual coverage (as our Definition 3) (ii) make no assumptions on the underlying distribution (iii) has finite-sample guarantee and asymptotic consistency. Such a triangle is shown by Vovk (2012) and Barber et al. (2021) to be impossible to achieve simultaneously. The listed works as well as ours follow different ways to reconcile the impossibility:

Paper 1 and Paper 5 focus on marginal coverage (as we define in our Definition 1), which greatly relaxes (i). Paper 3 and Paper 4 consider some milder violations of (i) but in different ways. Paper 3 considers a “clustered conditional coverage”, which is parallel to the group calibration defined in our Definition 2. Paper 4 considers another midway between marginal coverage and conditional coverage. Yet, these two papers are still far from conditional coverage (i).

In short, all these 4 papers have relaxed requirement (i). But what’s the cost to reach such individual coverage theoretically and what’s the price of keeping the requirement (i)? These questions are not answered by these papers. Our paper answers them with minimal relaxations on the other two: we only relax the “no distribution assumption” requirement (ii) with our Assumption 1. In comparison, Paper 2 also doesn't relax requirement (i) and is the most related work to ours. However, it relaxes both (ii) and (iii), and it violates (ii) to a further extent than ours. To see it, the algorithm in Paper 2 is based on a regularized pinball loss/ interval score loss, where the regularizer forces the covering interval length to be independent of the coverage indicator. It achieves a theoretical guarantee that the true conditional quantile is the minimizer of the regularized population loss under (1) a realizability assumption (that is, the true conditional quantiles lie in the function class considered) and (2) an assumption that the conditional distribution of $Y$ on $X=x$ is continuous with respect to $x$.

Apart from stronger assumptions, Paper 2's theoretical results have several limitations. First, it achieves only asymptotic consistency compared to our finite-sample guarantee in Theorem 1, which means that Paper 2 violates requirement (iii). Besides, it only achieves a necessary condition that the true conditional quantile is the minimizer of the population loss but not vice versa. Second, Paper 2 violates (ii) to a much deeper content than ours. Note that the key assumptions made in Paper 2 are the zero-approximation-error assumption and the continuity assumption. The former is not a weak one in the statistical learning theory, meaning that there should be no model misspecification at all, which limits the guarantee from a desirable “distribution-free” property. On the contrary, our algorithm can have a finite-sample guarantee with only three very general assumptions. Even for the continuity part of which the reviewer is unsatisfied, our assumption can still be relaxed to a weaker one than Paper 2 to reach the same asymptotic result. To see that, our result only requires the conditional quantile to be continuous, while Paper 2 requires continuity for the whole conditional distribution. Third, the regularization term in Paper 2 is related to a zero-one indicator, which is discontinuous for gradient descent. The authors twist their algorithm by replacing it with a smooth approximation to overcome the issue, without any formal guarantee on the twisted one. In contrast, our theory accompanies the exact same algorithm that we propose.

In short, many existing works on calibration completely ignore works on conformal prediction. We didn’t do this in our paper; in contrast, we’d like to call for more mutual awareness between these communities. Also, while we position our work along with the calibration literature, our theoretical analysis complements the existing literature on conformal prediction.

---

### Decision · Program_Chairs · 2023-09-21

**Decision:**

Accept (poster)

**Comment:**

The paper has received mostly favorable reviews, with the exception of reviewer bxwF, who has noted the paper is missing a comparison to work in conformal prediction, and expressed concern over the proposed method appearing to suffer from the curse of dimensionality. The authors have provided extensive responses addressing these two issues. After additional deliberation and discussion with the reviewers, I believe that - once properly extended and incorporated in the paper - the author's responses sufficiently address these issues. Consequently, I recommend acceptance subject to the following changes in the camera-ready paper:

* The literature review must include detailed discussion of all pertinent conformal prediction literature. Following commentary by reviewer aPS1, I encourage the authors to streamline the introduction by separating the detailed discussion of related work into its own section or subsection.

* The experiments must include the conformal prediction discussed during the review process as baselines.

* The experiments must report prediction interval sizes among other metrics, and discuss the pros and cons of various metrics along the lines mentioned during the review process.

* The authors should include additional discussion on the prospects of applying the proposed method to high-dimensional models such as neural networks, and perhaps add experiments with applying the model on second-to-last layer embedding from pre-trained neural networks.